There are amendments to this paper

# Dynamic changes of muscle insulin sensitivity after metabolic surgery

Sofiya Gancheva[1,2,3,11], Meriem Ouni [3,4,11], Tomas Jelenik[2,3], Chrysi Koliaki[1,2,3,5], Julia Szendroedi[1,2,3], Frederico G.S. Toledo[6], Daniel F. Markgraf [2,3], Dominik H. Pesta [2,3], Lucia Mastrototaro[2,3], Elisabetta De Filippo[2,3], Christian Herder [1,2,3], Markus Jähnert [3,4], Jürgen Weiss[3,7], Klaus Strassburger [3,8], Matthias Schlensak[9], Annette Schürmann [3,4,10,12] & Michael Roden [1,2,3,12]

The mechanisms underlying improved insulin sensitivity after surgically-induced weight loss are still unclear. We monitored skeletal muscle metabolism in obese individuals before and over 52 weeks after metabolic surgery. Initial weight loss occurs in parallel with a decrease in muscle oxidative capacity and respiratory control ratio. Persistent elevation of intramyocellular lipid intermediates, likely resulting from unrestrained adipose tissue lipolysis, accompanies the lack of rapid changes in insulin sensitivity. Simultaneously, alterations in skeletal muscle expression of genes involved in calcium/lipid metabolism and mitochondrial function associate with subsequent distinct DNA methylation patterns at 52 weeks after surgery. Thus, initial unfavorable metabolic changes including insulin resistance of adipose tissue and skeletal muscle precede epigenetic modifications of genes involved in muscle energy metabolism and the long-term improvement of insulin sensitivity.

[1] Division of Endocrinology and Diabetology, Medical Faculty, Heinrich-Heine University, Düsseldorf, Germany. [2] Institute for Clinical Diabetology, German Diabetes Center, Leibniz Center for Diabetes Research, Heinrich Heine University, Düsseldorf, Germany. [3] German Center for Diabetes Research (DZD e.V.), Neuherberg, Germany. [4] German Institute of Human Nutrition, Department of Experimental Diabetology, German Institute of Human Nutrition, Potsdam-Rehbruecke, Germany. [5] Laikon University Hospital, Athens, Greece. [6] Division of Endocrinology and Metabolism, Department of Medicine, University of Pittsburgh School of Medicine, Pittsburgh, PA, USA. [7] Institute for Clinical Biochemistry and Pathobiochemistry, German Diabetes Center, Leibniz Center for Diabetes Research, Heinrich Heine University, Düsseldorf, Germany. [8] Institute for Biometrics and Epidemiology, German Diabetes Center, Leibniz Center for Diabetes Research, Heinrich Heine University, Düsseldorf, Germany. [9] General Surgery Department, Schön Clinics, Düsseldorf, Germany. [10] Institute of Nutritional Sciences, University of Potsdam, Nuthetal, Germany. [11] These authors contributed equally: Sofiya Gancheva, Meriem Ouni. [12] These authors jointly supervised this work: Annette Schürmann, Michael Roden. Correspondence and requests for materials should be addressed to M.R. (email: michael.roden@ddz.uni-duesseldorf.de)

Obesity-related insulin resistance associates with greater availability of free fatty acids (FFA), mitochondrial alterations and inflammation[1,2]. Bariatric or metabolic surgery improves whole body insulin sensitivity on the long term[3], while the early metabolic effects and underlying mechanisms remain elusive[4–7].

Recently, we showed upregulation of both hepatic mitochondrial capacity and oxidative stress in obese individuals undergoing metabolic surgery[8]. Skeletal muscle mainly accounts for whole body insulin sensitivity and insulin resistant humans generally exhibit reduced muscle mitochondrial capacity[1,9]. Recent studies convincingly demonstrated that diet-induced weight loss rapidly improves hepatic, but not muscle insulin resistance in humans[10]. Metabolic surgery seems to exert variable effects on mitochondria[11,12] and may also affect lipolysis[4] and intracellular lipid mediators, but the time course of effects on skeletal muscle is unclear.

Gastric bypass surgery results in epigenetic alterations, but current data on DNA methylation are conflicting[13]. Specifically, it is unknown whether epigenetic alterations occur very early or rather late after metabolic surgery and relate to gene expression.

Here, we report the distinct epigenetic, transcriptional and metabolic changes in skeletal muscle and their dynamic temporal relationships during the improvement of insulin sensitivity by combined monitoring of systemic and cellular metabolism after metabolic surgery in obese humans (OB) and by comparing the pattern with that of healthy nonobese humans (CON).

## Results

**Obese humans feature higher muscle lipid intermediates.** Before surgery (baseline), OB had higher fasting glucose, insulin, ultrasensitive CRP (usCRP), and interleukin-6 (IL-6) than CON (Table 1). Insulin sensitivity at the level of adipose tissue (Adipo-IR) assessed from fasting FFA and insulin concentrations (Fig. 1a) and of whole body/skeletal muscle (M-value assessed from hyperinsulinemic-euglycemic clamps at steady state insulinemia of $58 \pm 14$ μU/ml; Fig. 1b) was lower in OB than in CON. OB also exhibited impaired metabolic flexibility ($\Delta$RQ) (Table 1).

Insulin resistance can result from accumulation of lipid mediators such as diacylglycerol (DAG), which activate novel protein kinase C (PKC) isoforms, or from sphingolipids, which

stimulate c-Jun N-terminal kinase (JNK)[14,15]. In muscle of OB, specific DAG species were increased in membranes (18:1 18:1), lipid droplets (18:1 18:1, 18:2 18:2 and 16:0 18:1) and cytosol (16:0 18:1) (Fig. 1c, Suppl. Fig. 1a, c). Also, membrane/cytosolic ratios were markedly elevated for PKCε ($2.1 \pm 1.9$ AU vs $0.4 \pm 0.3$ AU in CON, $p = 0.01$ using unpaired t-test) and there was a trend towards higher PKCθ ($p = 0.227$ using unpaired t-test, Fig. 1d) indicating enzyme activation. However, neither ceramides nor the Thr183/Tyr185-phosphorylated JNK ratios were different between OB and CON at baseline (Fig. 1e, f, Suppl. Fig. 1b, d, f). High-resolution respirometry of vastus lateralis muscle revealed lower maximal uncoupled respiration, when expressed per mg tissue (Fig. 2a), but unchanged citrate synthase activity (CSA) in OB (Fig. 2b). Electron transport chain (ETC) complexes II and III were lower in OB, while I, IV and V were similar in both groups (Fig. 2c, Suppl. Fig. 2a–d). Respiratory control ratio was lower, while leak control ratio was similar in OB vs. CON (Fig. 2d, e). Of note, muscle mitofusin-2 (Mfn2) and optic atrophy type 1 (Opa1) were lower in OB at baseline indicating reduced mitochondrial fusion activity (Suppl. Fig. 3a, b), while the autophagy markers, microtubule-associated protein 1A/1B-light chain 3 (LC3: $0.42 \pm 0.59$ vs $0.52 \pm 0.20$ in CON, $p = 0.22$ using unpaired t-test) and p62 protein (sequestosome 1, p62: $2.14 \pm 0.69$ vs $1.50 \pm 0.58$ in CON, $p = 0.17$ using unpaired t-test) were comparable between groups. Also, serum oxidation-reduction potential, antioxidant capacity and lipid peroxidation from thiobarbituric acid reactive species (TBARS) were not different (Fig. 2f, Table 1).

**Transient decrease in adipose tissue insulin sensitivity.** Forty-nine OB were studied at 2, 12, 24, and 52 weeks after surgery. Surprisingly, despite rapid weight loss of $10 \pm 3$ kg ($6.8 \pm 1.6\%$ of body weight) within 2 weeks, fasting glucose, insulin, C-peptide and inflammatory markers remained unchanged (Table 1). Insulin sensitivity did not improve in skeletal muscle and even deteriorated in adipose tissue (Fig. 1a, b). Of note, adipose tissue insulin resistance (Adipo-IR) remained ≈3-fold higher also at 12 and 24 weeks after surgery. The latter abnormality results from a rise in plasma FFA concentrations by 56% at 2 weeks compared to baseline (Table 1). The higher Adipo-IR occurred in the presence of marked increases in several muscle membrane and lipid

### Table 1 Participants' characteristics

| Parameter | CON | OB Baseline | OB 2 w | OB 12 w | OB 24 w | OB 52 w |
|---|---|---|---|---|---|---|
| N (male) | 14 (9) | 49 (14) | 42 (13) | 47 (13) | 49 (14) | 45 (12) |
| Age (years) | 40.3 ± 7.3 | 40.4 ± 10.0 | | | | |
| BMI (kg/m²) | 24.5 ± 3.7 | 51.4 ± 7.1# | 47.6 ± 6.8* | 42.3 ± 6.6* | 38.5 ± 6.6* | 33.9 ± 6.1* |
| Body weight (kg) | 75 ± 18 | 154 ± 27# | 143 ± 25* | 127 ± 24* | 115 ± 23* | 101 ± 20* |
| Change in BW (%) | – | – | 6.8 ± 1.6 | 17.4 ± 3.7 | 25.3 ± 5.5 | 33.0 ± 7.7 |
| Glucose (mg/dl) | 79 ± 8 | 98 ± 24# | 93 ± 26 | 84 ± 17* | 83 ± 14* | 80 ± 12* |
| Insulin (μU/ml) | 6(3;8) | 21(18;29)# | 22(14;29) | 12(8;17)* | 10(7;14)* | 9(5;11)* |
| C-peptide (ng/ml) | 1.3(1.1;1.6) | 3.3(2.7;4.7)# | 3.8(2.4;4.5) | 2.4(1.9;3.0)* | 2.1(1.6;2.9)* | 1.8(1.5;2.5)* |
| HbA1c (%) | 5.2 ± 0.3 | 5.8 ± 0.8# | 5.5 ± 0.8* | 5.3 ± 0.5* | 5.2 ± 0.5* | 5.2 ± 0.4* |
| FFA (μmol/l) | 492 ± 275 | 676 ± 161 | 1057 ± 263* | 699 ± 220 | 643 ± 239 | 527 ± 221* |
| Triglycerides (mg/dl) | 843(69;133) | 131(86;177) | 113(86;139) | 106(87;131)* | 97(75;130)* | 92(72;113)* |
| usCRP (mg/dl) | 0.1(0.1;0.2) | 0.7(0.4;1.3)# | 0.6(0.4;1.2) | 0.5(0.3;0.7)* | 0.3(0.2;0.7)* | 0.1(0.1;0.3)* |
| IL-6 (pg/ml) | 1.0(0.9;1.3) | 3.6(2.4;4.7)# | 3.0(2.1;4.4) | 2.4(2.2;3.5)* | 2.6(2.1;4.1) | 1.6(1.0;2.0)* |
| HMW-adiponectin (ng/ml) | 3181 (2491;4362) | 1432 (1065;3069)# | 2170 (1327;3346)* | 2708 (1579;3394)* | 3217 (1766;4211)* | 4025 (3118;6709)* |
| TBARS (μmol/mg protein) | 13(11;19) | 11(8;16) | 11(7;18) | 8(6;14) | 8(6;10)* | 6(5;8)* |
| Static ORP (mV) | 169 ± 11 | 162 ± 12 | 160 ± 8 | 161 ± 9 | 161 ± 8 | 164 ± 10 |
| REE (kcal/d) | 1550 (1346;1686) | 2240 (2026;2692)# | 1928 (1756;2201)* | 1875 (1641;2067)* | 1804 (1580;2065)* | 1815 (1558;2015)* |
| ΔRQ | 0.12 ± 0.03 | 0.07 ± 0.07# | 0.05 ± 0.08 | 0.08 ± 0.05 | 0.12 ± 0.06* | 0.16 ± 0.08* |

Mean ± SD or median(q1;q3)
CON lean humans, OB obese humans, BW body weight, TG-triglycerides, FFA free fatty acids, HMW adiponectin-high molecular weight adiponectin, TBARS thiobarbituric acid reactive substances, ORP oxidation-reduction potential, REE resting energy expenditure, RQ respiratory quotient, ΔRQ RQ_clamp – RQ_baseline
*$p < 0.05$ vs OB baseline using CPM for repeated measures analysis, #$p < 0.05$ vs CON using unpaired two-tailed t-test

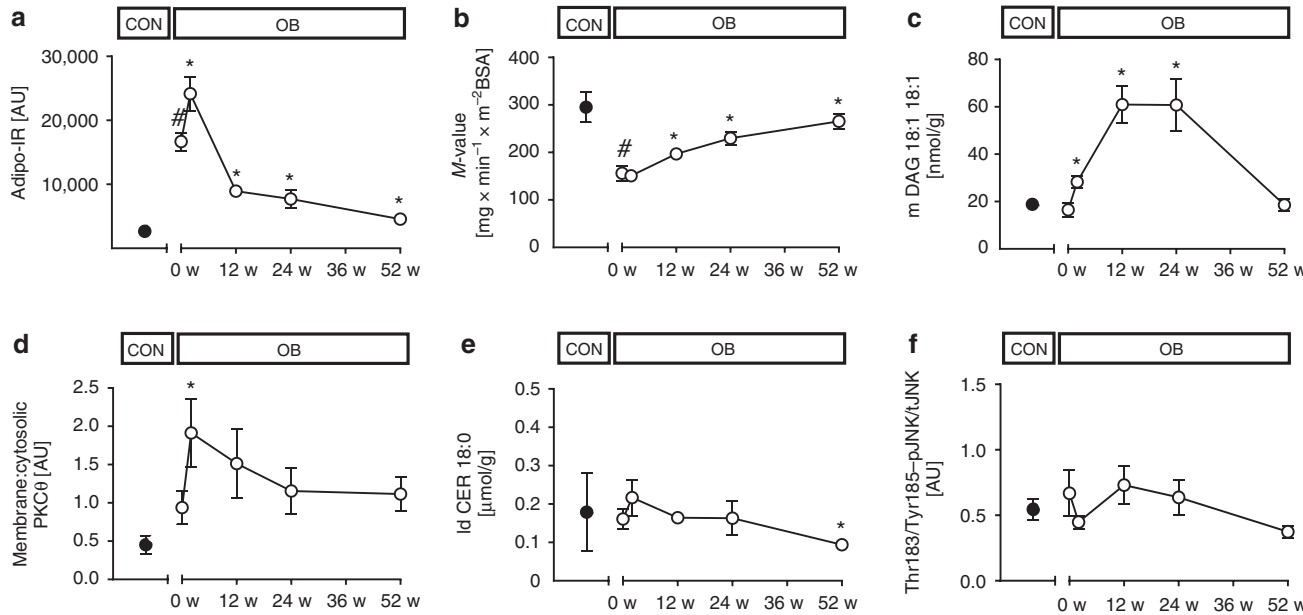

**Fig. 1** Time course of metabolic changes. Time course of changes in adipose tissue insulin sensitivity given as Adipo-IR (fasting free fatty acids*fasting insulin) (CON $n = 14$, OB $n = 47$) (**a**), muscle insulin sensitivity (CON $n = 7$, OB $n = 43$) (**b**), muscle membrane DAG 18:1 18:1 (CON $n = 4$, OB $n = 10$) (**c**), muscle protein kinase C (PKC) θ activation (CON $n = 4$, OB $n = 9$) (**d**), muscle lipid droplet ceramide 18:0 (CON $n = 4$, OB $n = 10$) (**e**) and muscle pJNK/ tJNK ratio (CON $n = 4$, OB $n = 8$) (**f**) in obese (empty circles) and in nonobese humans at baseline (black circles). Mean ± SEM, *$p < 0.05$ vs OB at baseline (0 w) using CPM for repeated measures analysis, #$p < 0.05$ vs CON using unpaired two-tailed $t$-test, CON nonobese humans, OB obese participants, BSA body surface area, LD lipid droplet, DAG diacylglycerol, pJNK phosphorylated c-Jun N-terminal kinase, tJNK total c-Jun N-terminal kinase

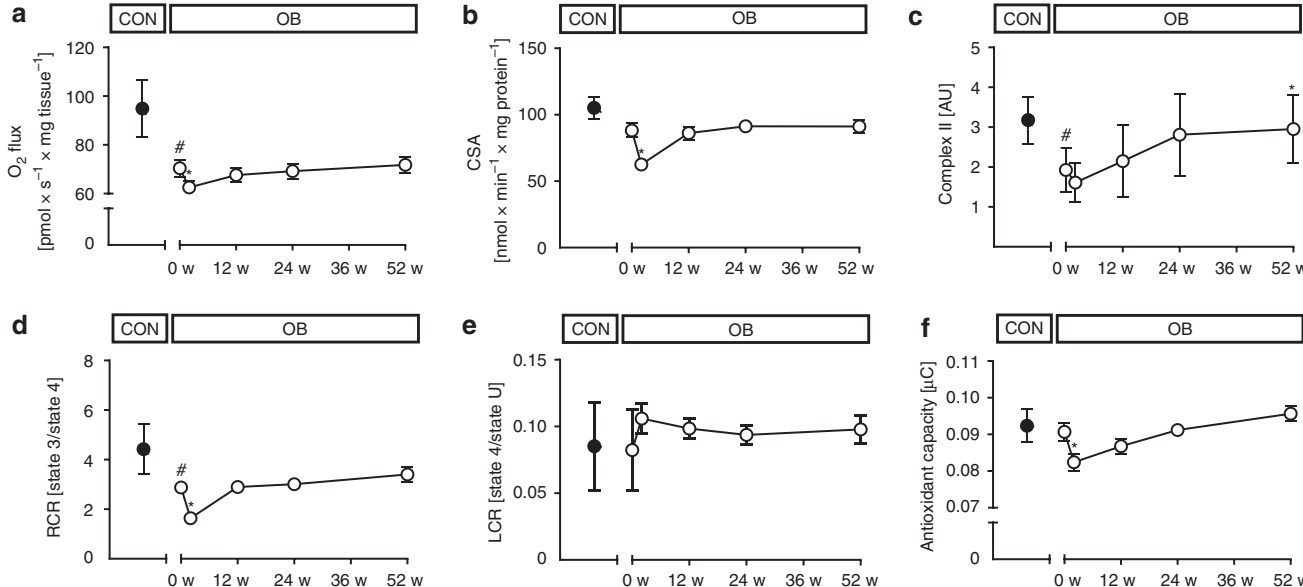

**Fig. 2** Time course of muscle changes. Time course of changes in muscle maximum uncoupled respiration (CON $n = 14$, OB n = 45) (**a**), citrate synthase activity (CSA) (CON $n = 13$, OB $n = 45$) (**b**), electron transport chain complex II succinate dehydrogenase complex iron sulfur subunit B protein content (CON $n = 10$, OB $n = 15$) (**c**), respiratory control ratio (RCR) (CON $n = 13$, OB $n = 43$) (**d**), leak control ratio (LCR) (CON $n = 13$, OB $n = 44$) (**e**) and serum antioxidant capacity (CON $n = 6$, OB $n = 28$) (**f**) in obese (empty circles) and nonobese humans at baseline (black circles). Mean ± SEM, *$p < 0.05$ vs OB at baseline (0 w) using CPM for repeated measures analysis, #$p < 0.05$ vs CON using unpaired two-tailed $t$-test. CON nonobese humans, OB obese participants

droplet DAG species (Fig. 1c, Suppl. Fig. 1a, e). Muscle PKCθ activation doubled ($p < 0.05$ using a covariance pattern model for repeated measures analysis, CPM; Fig. 1d) and PKCε also showed a trend towards higher values ($1.5 ± 1.5$ vs $2.1 ± 1.9$ AU, $p = 0.16$, CPM) at 2 weeks. Membrane ceramide C18:1 and cytosolic ceramide 18:1 increased, while cytosolic and membrane ceramide 24:0 even decreased at 2 weeks (Suppl. Fig. 1b, d, f), all without

any changes in pJNK/tJNK (Fig. 1f). Maximum respiration per mg tissue decreased by 10% in the uncoupled state (Fig. 2a) and by 17% in state 3. At 2 weeks, measures of muscle mitochondrial content showed variable results with lower CSA and density from transmission electron microscopy (Fig. 2b, Suppl. Fig. 4), but unchanged ETC complexes I–IV (Fig. 2c, Suppl. Fig. 2a–d). Compared to baseline, ETC complex V increased already at

2 weeks (Suppl. Fig. 2d). This was paralleled by a 43% lower respiratory control ratio indicating impaired mitochondrial efficiency (Fig. 2d). At 2 weeks, muscle Mfn2, Opa1, Fis1, Pink1, phospho-Pink1, Parkin, phospho-Parkin, DRP-1, phospho-DRP1 (Suppl. Fig. 3) as well as LC3 (0.45 ± 0.34 vs. 0.42 ± 0.59 AU, $p = 0.26$ using CPM) and p62 did not differ from baseline (2.21 ± 0.69 vs. 1.50 ± 0.58 AU, $p = 0.15$, using CPM). Serum antioxidant capacity was also decreased, while oxidation-reduction capacity and lipid peroxidation remained unchanged at 2 weeks (Fig. 2f, Table 1).

**Metabolic abnormalities normalize at 52 weeks after surgery.**
At 52 weeks, the total weight loss of 50 ± 14 kg (33.0 ± 7.7% of initial body weight) was accompanied by progressive improvements in glycemia and insulinemia (Table 1). A subgroup comparison between patients undergoing gastric bypass and sleeve gastrectomy surgery did not suggest differences in the time course of changes in body weight and muscle insulin sensitivity. However, the relatively small subgroup size does not allow to draw firm conclusions on a possible metabolic difference between these surgical techniques. The transiently increased plasma FFA improved by 22% (Table 1) and adipose tissue insulin resistance (Adipo-IR) gradually decreased until 52 weeks, but remained higher in OB than in CON (Fig. 1a). Muscle insulin sensitivity reached values of CON at 52 weeks (Fig. 1b). Muscle membrane and lipid droplet DAG continued to rise until 24 weeks before decreasing to baseline values at 52 weeks (Fig. 1c, Suppl. Fig. 1). Membrane/cytosolic PKCθ translocation decreased in parallel at 12 and 24 weeks towards baseline values (Fig. 1d). PKCε showed a trend towards lower values (1.19 ± 0.94 vs 2.14 ± 1.86 AU at baseline, $p = 0.053$ using CPM). Measures of mitochondrial function and content returned towards baseline (Fig. 2a–d, Suppl. Fig. 4). Compared to baseline, ETC complexes I, III and IV started to increase at 12 weeks, followed by II at 52 weeks (Fig. 2c, Suppl. Fig. 2a–d). Of note, Mfn2, Opa1, Fis1 (Suppl. Fig. 3a-c), LC3 (0.6 ± 0.4 vs. 0.4 ± 0.6 AU at baseline, $p = 0.03$ using CPM) and p62 (2.0 ± 0.5 vs. 1.5 ± 0.6 AU at baseline, $p = 0.01$ using CPM) were increased at 52 weeks. There was no difference in phospho-Pink1, phospho-Parkin and phospho-DRP1 at 52 weeks (Suppl. Fig. 3). Of note, inflammatory markers decreased, while high-molecular-weight (HMW) adiponectin rose at 12 and 24 weeks (Table 1).

**Rapid effects on gene expression after metabolic surgery.** To examine whether the metabolic differences reflect altered gene transcription, we initially compared the muscle transcriptome between OB before surgery (OB 0 w) and CON and detected 595 differentially expressed genes (Fig. 3a, Supplementary Data 1). Gene ontology (GO) analysis indicated that several genes are implicated in the regulation of apoptotic processes (GO:0042981), MAPK cascade (GO:0043410), inflammatory response (GO:0006954, GO:0050729), oxygen transport (GO:0015671) and others (Fig. 3b; Suppl. Table 3).

Comparing the muscle transcriptome of OB over the time course after surgery identified 937 out of 1,528 upregulated and 591 downregulated genes at 2 weeks (Fig. 3a, Supplementary Data 1). GO analysis showed significant enrichments in genes controlling transcriptional regulation (GO:0000398, GO:0010501), small GTPase-mediated signal transduction (GO:0051056) and negative regulation of insulin receptor signaling (GO:0046627) (Fig. 3c, Suppl. table 4). Among all transcripts, 1,244 were transiently altered at 2 weeks and then returned to baseline values (see also below), whereas 203 remained constantly changed (Fig. 3a). These transcripts comprise genes related to regulation of gene expression

(GO:0010467), calcium mediated signaling (GO:0019722) and negative regulation of insulin receptor signaling (GO:0046627) (Suppl. Fig. 5a, Suppl. table 5).

The genes exhibiting strongest changes at 2 weeks are involved in mitochondrial function (HMGCS2), lipid metabolism (ANGPTL4, ABCA1, ABCG1), calcium signaling (PCDH15), protein folding (DNAJC28), and inflammatory processes (CISH) (Suppl. Table 1). As altered mitochondrial functionality and lipid homeostasis were also the key metabolic abnormalities at 2 weeks (Figs. 1, 2) and because of the known association between abnormal mitochondrial, lipid and calcium homeostasis with insulin resistance, we further analyzed genes of these pathways over the entire post-surgical time period. This revealed 70 mitochondrial, 23 lipid metabolism and 60 calcium-related genes with significantly different expression levels (Fig. 3d). Importantly, several transiently regulated mitochondrial genes are coding for ribosomal RNA and tRNA or are known to modify mitochondrial tRNAs[16] suggesting that the mitochondrial transcription and translation machinery is highly active immediately after surgery.

At 12 weeks after surgery, 625 genes were significantly altered (341 up- and 284 downregulated), at 24 weeks 756 genes (519 up- and 237 downregulated) in comparison to their expression before surgery (Fig. 3a, Supplementary Data 1). The pathways affected at 12 weeks comprise—among others—glycogen metabolic process (GO:0005977), translation (GO:0006413 and GO:0006364) and transmembrane transport (GO:0055085) (Suppl. Fig. 5b; Suppl. table. 6). At 24 weeks, genes related to transmembrane receptor protein tyrosine kinase signaling (GO:0007169), cytoskeleton organization (GO:0007010) and regulation of cell growth (GO:0001558) are mostly higher expressed than before surgery (Suppl. table 7).

At 52 weeks, 1,449 transcripts were significantly altered (Fig. 3a, Supplementary Data 1), which comprise genes involved in glycogen metabolic process (GO:0005977), protein polyubiquitination / destabilization (GO:0016567, GO:0031648), (Suppl. table 8) and including upregulation of those contributing to transcription (BACH1), cytoskeletal and tubulin reorganization (MYH3, CCDC8, ACTC1) (Suppl. table 2 and 8).

**Alterations of DNA methylation occur later after surgery.**
Mitochondria are also essential for providing metabolites for the generation and modification of epigenetic marks, which in turn modulate gene expression[17]. Thus, we examined whether the transient increase in muscle mitochondrial oxidative capacity could contribute to reprogramming of DNA methylation. Indeed, 107,325 CpG sites showed differential muscle DNA methylation between OB and CON. The methylation of individual CpG sites before and after metabolic surgery was compared by paired analysis. At 2 weeks, only 19 CpGs were different, of which 14 remained altered at 52 weeks (Fig. 4a). At 52 weeks, 109,105 CpGs were differentially methylated compared to baseline (Fig. 4a). Of note, most differentially methylated CpG sites (89% at 2 weeks and 70% at 52 weeks) were hypomethylated after surgery (Suppl. Fig. 6a, b). Changes in DNA methylation were very low (between −5 and +4%) at 2 weeks, and higher (−14 to +11%) at 52 weeks when compared to baseline. The number of differentially methylated CpG sites at 2 weeks was low and not enriched in a specific genomic region (Suppl. Fig. 6c). Changes in DNA methylation observed at 52 weeks occurred mostly outside CpG islands, particularly in open sea and shelf areas ($p < 2 \times 10^{-16}$, $p = 2.3 \times 10^{-8}$ Pearson's Chi-squared test). A significant enrichment was visible in intergenic regions and gene bodies at 52 weeks ($p < 2 \times 10^{-16}$, Chi-square test, Suppl. Fig. 6c). This enrichment explains the relatively small overlap between

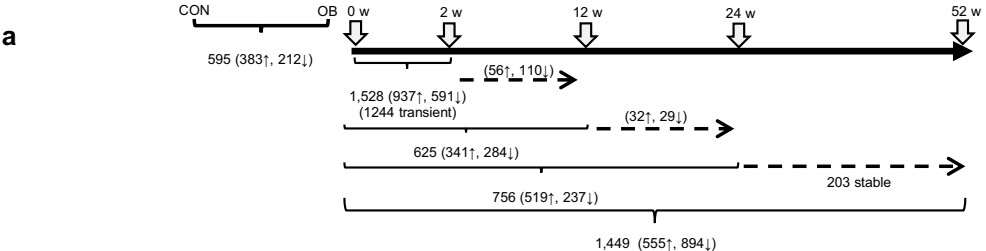

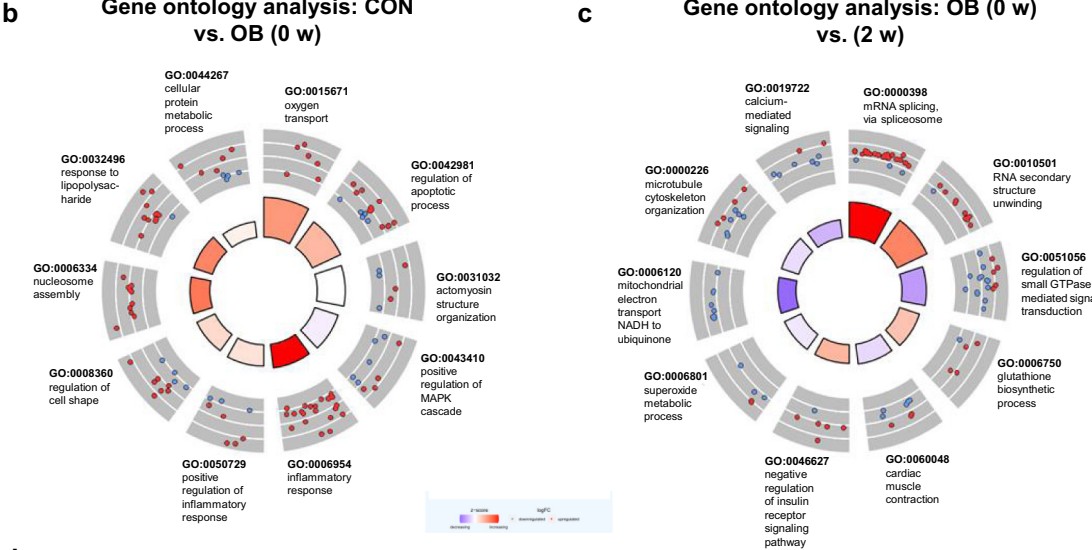

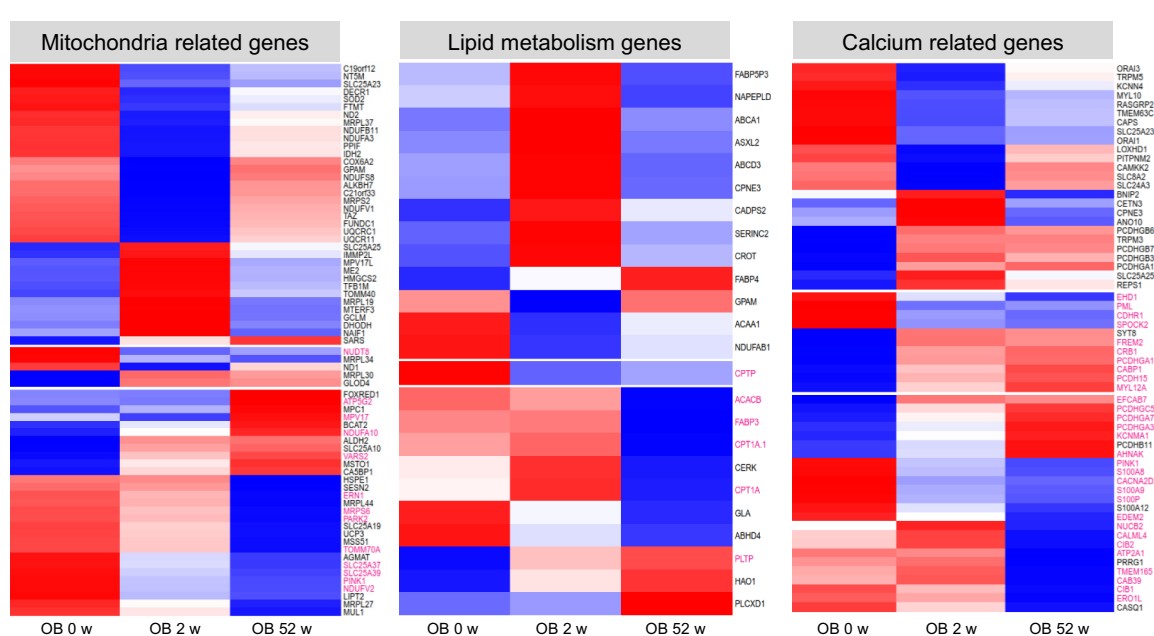

**Fig. 3** Differences in skeletal muscle transcriptome. Transcriptome analysis of skeletal muscle before (0 weeks), at 2, 12, 24 and 52 weeks after metabolic surgery. Data are given as number of differentially expressed transcripts between indicated groups and time points (**a**). Gene ontology analysis of genes differentially expressed between lean (CON) and obese (OB) participants. The inner circle depicts the main processes to be increased (blue) or decreased (red) in OB. The outer circle shows scaled scatter plots for affected genes and their regulation within the most-enriched biological pathway in OB at baseline (OB, 0 w) (**b**) and 2 weeks after the surgery (OB, 2 w) (**c**). Changes in mRNA expression of genes related to mitochondrial function, lipid metabolism and calcium signaling in skeletal muscle (**d**). Heat maps indicate expression levels of listed genes at 0 (baseline), 2 and 52 weeks after metabolic surgery. Each column represents the average expression level of 16 individuals and each row shows the expression profile of one single transcript with significant differences. Up- and downregulated genes are indicated by red and blue signals, respectively; the signal intensity corresponds to the log-transformed magnitude of the average of expression per group. All genes printed in bold magenta show differentially methylated CpGs at 52 weeks. For gene expression unadjusted p-value and DNA methylation data are adjusted for multiple testing with Benjamini Hochberg correction *$p < 0.05$ (unpaired (CON vs OB) and paired (OB 0 vs 2 weeks/52 weeks two-tailed $t$-test; CON: $n = 6$, OB: $n = 16$)

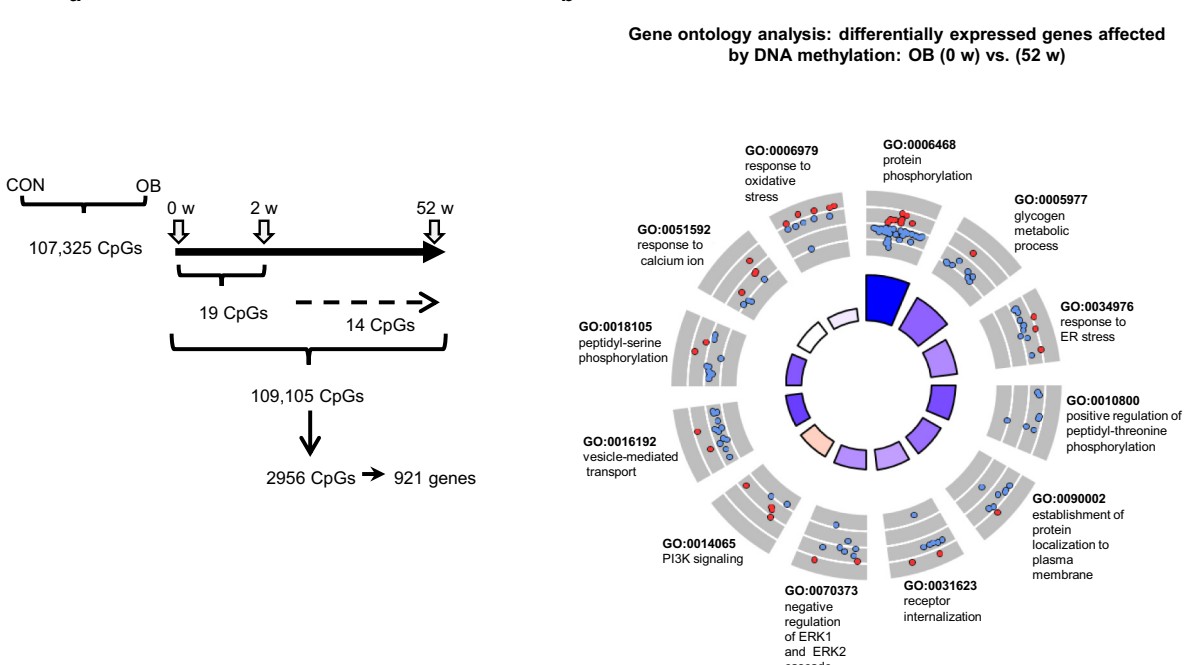

**Fig. 4** Genome wide DNA methylation analysis in skeletal muscle. The number of differentially methylated CpG sites between the indicated groups (**a**). A total of 2956 CpG sites are located in/or in close proximity of 921 genes exhibiting different expression between baseline and 52 weeks. Gene ontology analysis of 921 differentially expressed genes with altered levels of DNA methylation at 52 weeks (**b**). Up- and downregulated genes are indicated by red and blue signals, respectively. *$p < 0.05$ (unadjusted $p$-value paired $t$ test for gene expression, $n = 16$, methylation data with Benjamini Hochberg correction)

differentially methylated CpGs and differentially expressed genes, as only 2956 CpGs are located in/or close to regions of the 921 differentially expressed genes (Fig. 4a). Among differentially expressed genes involved in mitochondrial function, lipid metabolism and calcium signaling, we detected 13, 6 and 30, respectively, which were affected by DNA methylation at 52 weeks (Fig. 3d, highlighted in magenta).

**Changes in gene expression via altered DNA methylation.** DNA methylation in promoter and enhancer regions is associated with silencing of gene transcription, whereas elevated DNA methylation is typically found in the body of actively transcribed genes[18]. To further investigate the relationship between DNA methylation and gene expression, we compared all upregulated genes with hypomethylated CpGs in promoters and hypermethylated CpGs in gene bodies and vice versa for the downregulated genes. None of the differentially expressed genes detected at 2 weeks showed altered DNA methylation. However, at 52 weeks, 2956 CpGs were identified in 921 differentially expressed genes (Fig. 4a). GO analysis indicated enrichment in genes linked to several biological pathways, including protein phosphorylation, glycogen metabolic process (GO:0005977), protein localization to plasma membrane and vesicle-mediated transport (Fig. 4b, Suppl. table 9). Among those, 323 contain >2 differentially methylated CpGs and levels of altered DNA methylation higher than 5 %. Only few genes (~20) were previously described to be epigenetically regulated in obese humans[13,19]. Taking genes with an altered expression and DNA methylation (>5%) of at least 2 CpG sites into account, which can also be linked to insulin sensitivity, we identified 94 gene candidates to be affected by epigenetic alterations (Suppl. Fig. 7). Examples are *PTPRE*, a negative regulator of insulin receptor (IR) signaling in skeletal muscle; *PIK3R1*, a regulator of the PI3-kinase; *MLXIP*, involved in transcriptional activation of glycolytic target genes, and *ACACB*, a mitochondrial enzyme playing a role

in fatty acid metabolism (Fig. 5a–d). A hypermethylation in the promoter region of *PTPRE* gene related to lower expression at 52 weeks (Fig. 5a). A lower DNA methylation in the promoter and a higher methylation in the gene body of *PIK3R1* is linked to its higher expression after the surgery (Fig. 5b), whereas a hypomethylation in the gene body of *MLXIP* and *ACACB* associated with their lower expression (Fig. 5c, d). In addition, the time course of the expression of the listed candidates was evaluated. *PTPRE* expression decreased already at 2 weeks, whereas the changes of the expression of *PI3KR1* occurred at 12 weeks and *ACACB* and *MLXIP* at 52 weeks (Suppl. Fig. 9).

**Correlation of altered expression and methylation.** In order to relate the differential expression and changes in DNA methylation to the primary metabolic phenotypes, e.g., body weight, insulin sensitivity, we calculated their correlation. Table 2 lists the expression levels of several genes associated with changes in *M*-value (27), fasting glucose (1219), HMW-adiponectin (73) and mitochondrial content (29). GO analysis of the 1219 affected genes, which correlated to glucose concentrations, can be linked to cytoskeleton organization (GO:0007010), calcium signaling (GO:0016338) and others (Suppl. table 10). The correlation analysis of the 921 differentially expressed and methylated genes (at 52 weeks) revealed 177 genes associated with BMI, 443 with *M*-value and 70 with HMW-adiponectin (Table 3). The correlations between methylation levels and M-value of the OB group for *PTPRE* ($R^2 = 0.286$; $p = 10^{-4}$, Pearson correlation) and *PIK3R1* ($R^2 = 0.312$; $p = 5.10^{-5}$, Pearson correlation) indicate that their interindividual epigenetic alterations are linked to the improvement in insulin sensitivity (Suppl. Fig. 10).

**Epigenetic reprogramming of transiently altered expression.** As 1150 mRNAs (encoded by 1126 genes) were only transiently differentially expressed at 2 weeks (Fig. 3a) and returned to

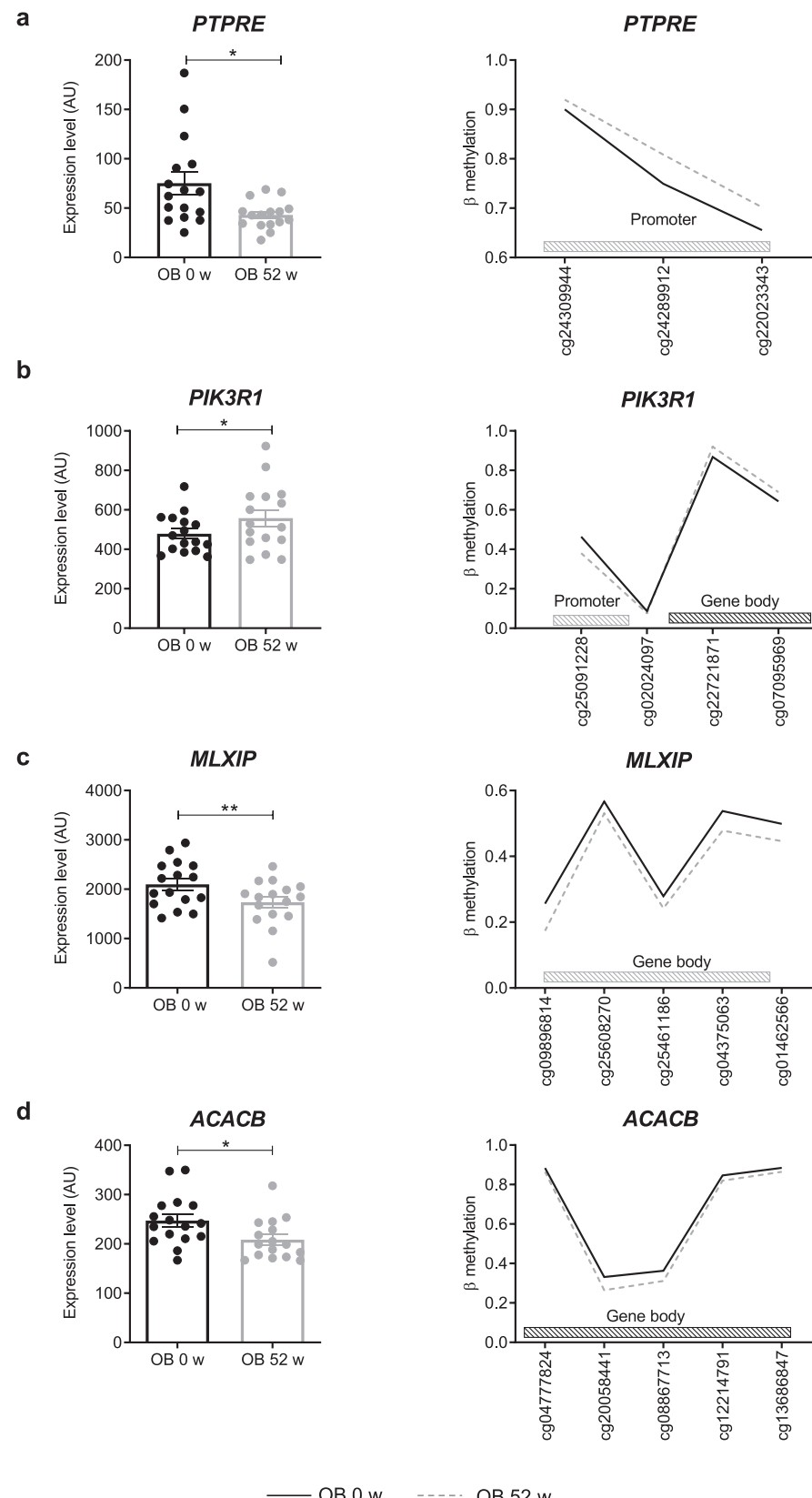

**Fig. 5** Gene candidates showing differences in expression and DNA methylation. Expression is shown in left panels and levels of DNA methylation in right panels. Differentially methylated CpGs are located at different positions of the genes, either in the promoter as shown for *PTPRE* (**a**), in the gene body as shown for *MLXIP* (**c**) and *ACACB* (**d**), or in both as depicted in *PIK3R1* (**b**). Mean ± SEM (left panels). Obese humans given as black circles/lines at baseline and as gray circles/lines at 52 weeks. Only significantly differentially methylated CpGs are represented; *p < 0.05 (Gene expression unadjusted *p*-value paired *t* test, *n* = 16, methylation data with Benjamini Hochberg correction)

| Table 2 Pearson correlations between gene expression levels and indicated clinical parameters | | | | | | |
|---|---|---|---|---|---|---|
|  | Body mass index | M-value | Fasting glucose | FFA suppression | HMW-adiponectin | Mitochondrial content (CSA) |
| Number of genes | 25 | 27 | 1219 | 76 | 73 | 29 |

| Table 3 Number of genes differentially expressed and methylated at 52 and exhibiting at least one CpG significantly correlated to the indicated clinical parameters | | | | | | |
|---|---|---|---|---|---|---|
|  | Body mass index | M-value | Fasting glucose | FFA suppression | HMW-adiponectin | Mitochondrial content (CSA) |
| Number of genes | 177 | 443 | 163 | 112 | 70 | 27 |

baseline expression levels, we tested whether epigenetic alterations were responsible for this effect. We therefore mapped the differentially methylated CpGs at 52 weeks to all transiently altered transcripts at 2 weeks. Indeed, 75% (849) of these genes showed changes in DNA methylation at 52 weeks (Fig. 6a; Chi-square $p < 10^{-255}$), including genes involved in mitochondrial function ($n = 12$), calcium signaling ($n = 11$), lipid metabolism ($n = 4$). Representative examples comprise *HMGCS2* (hydroxymethylglutaryl-CoA synthase[20], and *IMMP2L* a mitochondrial inner membrane protease subunit 2 involved in peptides translocation into mitochondria[21,22] (Fig. 6b; Suppl. Fig. 9). Collectively, these data indicate that changes in methylation at 52 weeks associate with the reversal of altered expression of genes involved in mitochondrial functionality and lipid metabolism.

## Discussion

This study provides insights into the mechanisms, by which metabolic surgery sequentially affects systemic and tissue-specific metabolism and its epigenetic regulation in obese humans. In particular, surgically-induced weight loss (i) is not immediately followed by improved muscle insulin resistance, likely related to transiently augmented lipolysis resulting in accumulation of lipid intermediates, inadequate mitochondrial function and altered gene expression profiles, (ii) may subsequently modify DNA methylation of genes involved in muscle energy metabolism, and (iii) associate with changes in gene expression along with restoration of muscle metabolism within one year. We thereby provide evidence for a role of transient changes in the expression of specific genes, which may be reprogrammed by DNA methylation.

Generally, the impressive weight loss by metabolic surgery is believed to be responsible for the improvement of whole-body metabolism, and specifically of insulin sensitivity leading to remission of type 2 diabetes[3,5,6]. Surprisingly, this study demonstrates that the rapid loss of ~10 kg equaling ~7% of body weight within 2 weeks, does not translate into immediate improvement of muscle insulin resistance. Interestingly, analysis of the few previous studies on early effects of metabolic surgery also found no[6,7] or minor increases in insulin sensitivity[4,23]. Using a low-caloric dietary intervention, Taylor and colleagues reported marked improvements in hepatic fat content and glucose production, but also no changes in muscle insulin sensitivity after one week[24,25]. The present study now identifies sustained elevation of adipose lipolysis as underlying cause, resulting in initial doubling of fasting plasma FFA with persisting adipose tissue insulin resistance for up to 6 months after surgery and parallel intracellular accumulation of 18:1 18:1 DAG species and PKCθ stimulation. Thereby, it shows that dynamic changes in endogenous FFA associate with activation of the DAG/PKC pathway in humans, which has been previously only demonstrated for high doses of exogenous lipid infusions[15,26]. The greater muscle PKCε activation in OB before surgery suggests that also this PKC isoform relates to obesity-mediated muscle insulin resistance as suggested[27] and reported for hepatic insulin resistance[28]. Other lipid mediators like ceramides, previously associated with human muscle insulin resistance in some[29], but not other studies[26,30], were generally not altered in muscle of OB. Membrane and cytosolic C18:1 ceramides were also elevated at 2 and 12 weeks, but without concomitant changes in muscle JNK phosphorylation, and subsequently rapidly decreased during follow-up as reported in lower-degree obesity[11]. Of note, several DAG as well as ceramide species were decreased, when insulin sensitivity had increased at 52 weeks. This occurred in parallel with a trend towards decreased PKCε activation and lower pJNK/tJNK at 52 weeks. The latter, however, could also be related to lower inflammatory activity[15] as supported by the lower usCRP and IL-6 levels. Taken together, these data indicate that the DAG-PKC pathway is primarily responsible for the initial lack of improvement in muscle insulin resistance upon metabolic surgery and along with reduced inflammatory activity underlies the later increase in insulin sensitivity.

Insulin resistance frequently, but not generally associates with impaired mitochondrial function[12,27,31–33]. The present study not only confirmed the lower muscle mitochondrial oxidative capacity and coupling efficiency and fusion activity in OB before surgery[32–34], but also provides evidence for a dynamic regulation of mitochondria in humans. At 2 weeks, mitochondrial mass decreased, as measured independently by both biochemical and ultrastructural analyses. Moreover, no changes in ETC complexes I–III were detected at 2 weeks, possibly suggesting a transient delay in the improvement of mitochondrial abundance. Intracellular lipid accumulation likely accounts for the reduction of mitochondrial mass, oxidative capacity and coupling efficiency early after surgery as reported for C2C12-differentiated myotubes[35]. Decreased antioxidant capacity could result from augmented ROS quenching in the presence of elevated lipid peroxidation and higher lipid influx. The continued impairment of mitochondrial fusion activity at 2 weeks, suggests that altered mitochondrial functionality persists during elevated lipid availability and only reverses with improved insulin action at 52 weeks[36]. Paralleled by decreasing plasma FFA, lower muscle mitochondrial function returned to baseline values within 12 weeks and then remained unchanged until 52 weeks, despite continuous improvement of insulin sensitivity in line with normalization of mitochondrial function in other studies with limited time resolution[12,37,38]. Our findings are in concert with long-term caloric restriction data demonstrating no change in mitochondrial oxidative capacity despite improved insulin sensitivity[39]. This indicates that surgical weight loss—similar to that by caloric restriction[40] or by inhibition of lipolysis[41]—dissociates modulation of insulin sensitivity from muscle mitochondrial respiration.

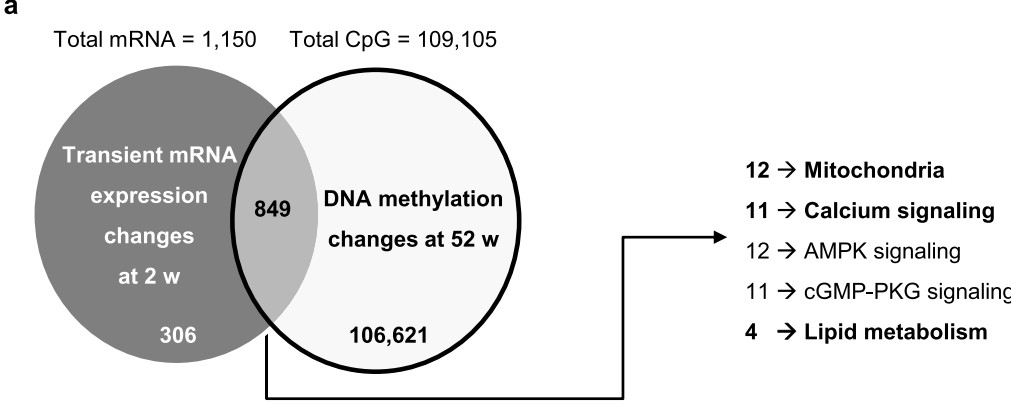

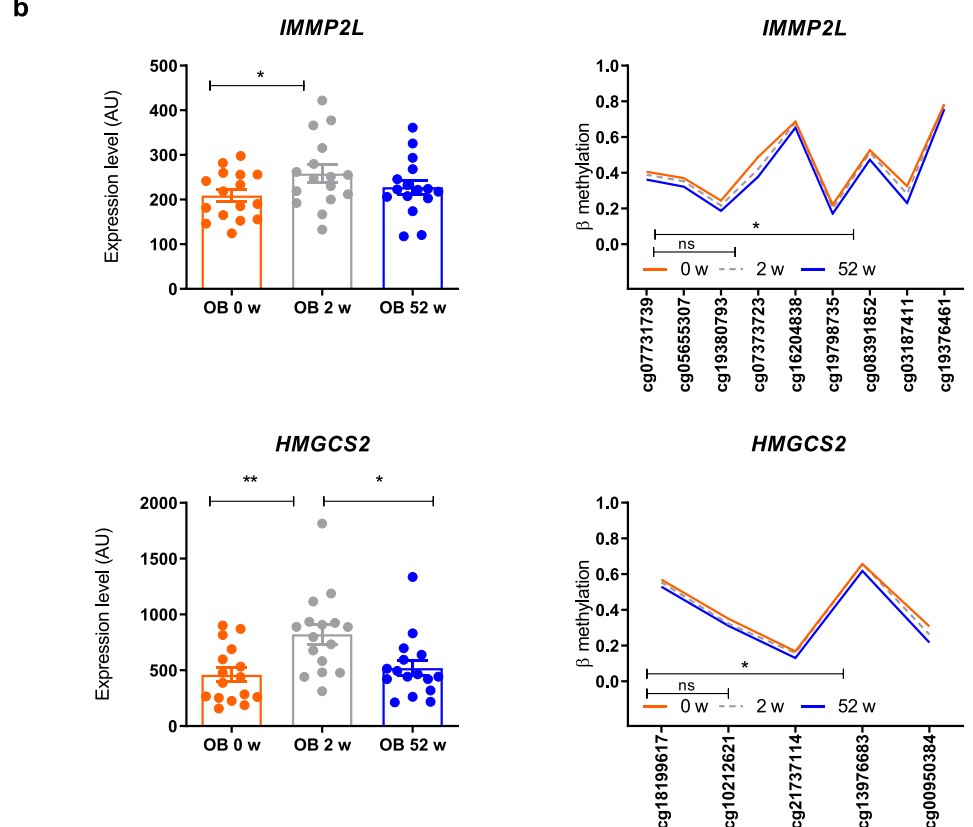

**Fig. 6** Reprogramming of gene transcripts by DNA methylation at 52 weeks. Seventy-five percent of genes exhibiting transient differences in expression at baseline show changes in DNA methylation at 52 weeks. Chi-square test $p < 10^{-255}$. Gene ontology analysis identified the indicated number of genes to be altered in the listed cellular processes (right panel) (**a**). Examples of changes in DNA methylation in *IMMP2L* (upper panel) and *HMGCS2* genes (lower panel). Both genes are only differentially expressed 2 weeks after the surgery (**b**). Mean ± SEM, obese humans are shown as red circles/lines at baseline, as gray circles/lines at 2 weeks and as blue circles/lines at 52 weeks after surgery. *$p < 0.05$ (Gene expression unadjusted $p$-value 2-tailed paired $t$ test, $n = 16$, methylation data with Benjamini Hochberg correction)

However, in contrast to our findings, short-term caloric restriction for 60 h or 53 days did not affect muscle mitochondrial density[42,43], suggesting acute intervention-specific effects of weight loss in humans. Nevertheless changes in diet and eating behavior, which have been extensively studied previously[44], were not assessed in this observational study, thereby limiting conclusions as to their possible confounding effect on weight loss. In addition to the changes resulting from surgery, moderate changes of measured variables could be also due to the repeated testing one year later.

Comprehensive analysis of muscle gene expression profiles reflected the observed metabolic alterations. Before surgery, as described previously[13,19] particularly genes involved in lipid metabolism, such as *FFAR4* (free fatty acid receptor 4), exhibited differential expression. At 2 weeks, the expression pattern mirrored the transient alterations of lipid metabolism and

mitochondrial function, which later returned to baseline or approached those of healthy humans. Elevated *ABCD3* (ATP-binding cassette sub-family D member 3) or *NAPELD* (N-acyl-phosphatidylethanolamine-hydrolyzing phospholipase D) expression at 2 weeks likely reflects the higher FFA uptake[45,46]. Also genes related to mitochondria and calcium handling showed higher expression levels, such as *SLC25A25* (calcium-binding mitochondrial carrier protein SCaMC-2), encoding a calcium-dependent mitochondrial solute carrier[47]. Of note, elevated mitochondrial $Ca^{2+}$ increases activities of glycerol 3-phosphate dehydrogenase, pyruvate dehydrogenase phosphatase, NAD-isocitrate dehydrogenase and 2-oxoglutarate dehydrogenase, all of which increase the NADH production for oxidative phosphorylation[48]. Another set of transiently upregulated genes relates to small GTPase-mediated signaling, such as *ARHGAP24, RACGAP1, ITSN1* and to negative regulation of insulin signaling, such as *SOCS1, RPS6KB1*. The latest mediates TNF-alpha-induced insulin resistance by phosphorylating IRS1 at multiple serine residues, resulting in accelerated degradation of IRS1[49–51]. This points to a role of local inflammatory processes as cause of adipose tissue insulin resistance leading to enhanced lipolysis early after surgery. At 52 weeks, downregulated genes comprise interferon-γ mediated signaling, in line with the time course of inflammatory markers of the present study and the lower expression of inflammation-related genes in a previous study[52]. Interestingly, interferon-γ has recently been closely linked to skeletal muscle insulin resistance and derailed glucose metabolism in obesity[53].

We also provide detailed description of time-dependent changes in muscle metabolism, transcriptome and methylome. At 2 weeks, several transient changes in skeletal muscle occurred in parallel, comprising augmented lipid availability, mitochondrial adaptation and altered expression of genes related to lipid metabolism and cAMP signaling. Interestingly, transcriptional changes at this time point are not linked to alterations in DNA methylation indicating that other regulatory mechanisms, e.g., activity of transcription factors, mediate the differences in gene expression. In contrast, at 52 weeks, many CpGs are differentially methylated, suggesting that alterations in the entire methylome may occur at a later stage[13]. However, in agreement with previous studies[13,54,55], changes in DNA methylation of ~5% are relatively small. The functional relevance of such moderate differences has been proven for some genes by reporter assays[56]. The improvement of mitochondrial flexibility after surgery-induced weight loss might provide substrates for epigenetic changes in muscle at 52 weeks. Interestingly, recent reports provided supporting information on the role of oxidative capacity for epigenetic changes[17,57,58]. Nearly 3000 of the differentially methylated CpGs are associated with altered expression of the corresponding 921 genes, pointing towards the relevance of epigenetic mechanisms in response to weight loss. Metabolic surgery remodels DNA methylation of genes involved in insulin signaling, such as *PIK3R1* and *PTPRE* and *ACACB* (also designated *ACC2*), which is essential for fatty acid metabolism[59]. Mice lacking *Acc2* exhibit higher fatty acid oxidation rates in the soleus muscle than control mice[60]. Previous studies found only three genes with differences in expression linked to changes in DNA methylation after surgery, *SORBS3* (sorbin and SH3 domain containing 3), *PGC-1α* (PPARG coactivator 1 alpha) and *PDK4* (pyruvate dehydrogenase kinase 4)[13]. The present study identified close to 300 epigenetic changes associated with altered expression of important metabolic genes, likely contributing to improved insulin sensitivity and lipid metabolism. Pathway enrichment analysis of the 921 differentially expressed and methylated genes indicates that several genes are implicated in the regulation of glycogen metabolism,

intracellular signal transduction and cAMP biosynthesis, reflecting improved skeletal muscle activity and metabolism[61].

Moreover, the present study found that changes in DNA methylation are associated with reprogramming up to 75% of the transiently altered genes, which possibly normalize their expression levels at 52 weeks (Fig. 6). Of note, genes undergoing such changes are involved in mitochondrial and lipid metabolism, again reflecting the early transient metabolic alterations. This data provides at least indirect evidence that DNA (de)methylation processes are possibly involved in the reprogramming of gene expression in response to metabolic surgery. The question is whether all changes in expression and DNA methylation are occurring in response to weight loss or if and to which extent weight-independent alterations contribute to observed phenotypes. The finding that the number of genes correlating with BMI is clearly lower than that of genes associating for instance with fasting glycemia or adiponectin levels indicates that not all observed effects are directly related to body weight loss (Tables 2 and 3). At present, no data are available on the time course of possible effects of a comparable diet-induced weight loss on epigenetic changes and alterations in energy metabolism. Consequently, the present data have to be interpreted in the context of surgical weight loss and cannot be generalized to weight loss by other interventions or causes. Taken together, surgically-induced weight loss induces several metabolic events leading to sustained improvement of insulin sensitivity. These effects involve transient dyslipidemia and lipotoxic insulin resistance as well as reduced mitochondrial mass associated with altered gene expression followed by long term metabolic reprogramming of gene expression possibly mediated by epigenetic mechanisms. These mechanisms underlie the dynamic changes of insulin sensitivity after bariatric surgery and point at mitochondrial function and lipid metabolism as key obesity treatment targets.

## Methods

**Study population.** We studied 49 obese Caucasian patients (OB) before and 2, 12, 24 and 52 weeks after sleeve gastrectomy or gastric bypass surgery (Suppl. Fig. 8). Healthy nonobese humans were examined once (controls, CON). The OB group included 13 patients with type 2 diabetes (T2D) presenting with good metabolic control (fasting glucose: $139 \pm 24$ vs. $92 \pm 16$ mg/dl, $p < 0.01$, unpaired two-sided $t$-test; HbA1c: $7.2 \pm 1.0$ vs. $5.6 \pm 0.5$ %, $p < 0.01$, unpaired two-sided $t$-test) and similar serum insulin, c-peptide and lipid levels as the glucose tolerant participants. T2D patients were drug naive ($n = 3$) or treated with monotherapy (metformin: $n = 3$, DPP-4 inhibitor: $n = 1$, GLP-1 agonist: $n = 1$) or combination therapy (metformin+insulin: $n = 2$, metformin+GLP-1 agonist: $n = 2$, metformin+GLP-1 +insulin: $n = 1$). Oral glucose-lowering medication and GLP-1 agonists were withdrawn 3 days, while long acting insulin analogues were replaced with NPH insulin for 7 days before the metabolic tests. If necessary, regular insulin was used during the night before the metabolic studies to avoid hyperglycemia. Participants were allowed to take thyroid hormones, oral contraceptives or antihypertensive treatment, but not immunomodulatory or other medications. All participants were non-smokers and engaged only in light physical activity and had no evidence for growth hormone deficiency, which has been shown to affect changes in lean mass and insulin sensitivity after bariatric surgery in other obese cohorts[62]. Six months prior to surgery patients underwent structured multimodal non-surgical weight-lowering treatment following the German guidelines for obesity management without achieving weight loss of >10%. Dietary and physical activity counselling was provided after the surgery according to local guidelines. Data of some participants were part of a previous report[8]. Each participant underwent 3-h hyper-insulinemic-euglycemic clamps using primed-continuous infusion of insulin (80 mU•m$^{-2}$•min$^{-1}$ for 8 min followed by 40 mU•m$^{-2}$•min$^{-1}$; Insuman Rapid, Sanofi, Frankfurt am Main, Germany) combined with indirect calorimetry for assessment of insulin sensitivity and metabolic flexibility from the change in the respiratory quotient (ΔRQ) during the clamp (RQ_steady state RQ_baseline)[8]. A variable infusion of 20% (w/v) glucose (B. Braun, Melsungen, Germany) was administered to maintain blood glucose at 90 mg/l, which was checked at 5-min intervals. Blood samples were collected before and during the clamp for measuring hormones and metabolites. All participants received information about all procedures and risks before providing their written consent to a protocol, approved by the ethics board of Heinrich-Heine University Düsseldorf (registered clinical trial, NCT01477957). The study was performed in compliance with all relevant ethical regulations for work with human participants.

**Mitochondrial content and function**. Mitochondrial respiration was assessed in permeabilized muscle fibers, obtained by biopsies from the vastus lateralis muscle[30]. Maximal oxidative phosphorylation (state 3) and resting respiration rates (state 4) were quantified upon sequential exposure to substrates followed by incremental titration steps of 1.0 μl carbonyl cyanide p-[trifluoromethoxy]-phenylhydrazone (FCCP) until maximal uncoupled respiration (state u) was achieved in a 2-chamber Oxygraph-2k (OROBOROS Instruments, Innsbruck, Austria)[41]. Respiratory rates were expressed per mg tissue weight as well as per individual CSA[1]. Respiratory control ratio (RCR), defined as state 3/state 4, was used as an index of mitochondrial coupling, while leaking control ratio (LCR) was calculated from state 4/state u ratio, as a marker of the proton leak. Mitochondrial content was quantified by transmission electron microscopy and stereological principles using a 144-point grid overlaid on micrographs[63,64]. For each biopsy, 10–12 micrographs (6300× and 25,000× magnification) of longitudinally-sectioned tissue were analyzed in a blinded fashion.

**Blood analyses**. Metabolites, insulin, C-peptide and hsCRP[33] as well as IL-6 and HMW adiponectin were assessed[65]. Serum concentrations of TBARS were measured fluorometrically (BioTek, Bad Friedrichshall, Germany)[66]. Plasma static oxidation-reduction potential (sORP) and antioxidant capacity were determined in plasma as markers of systemic oxidative stress using the RedoxSYS (Luoxis Diagnostics, Inc., Englewood, CO, USA)[66].

**Protein analysis**. Activities of PKCθ and PKCε were assessed from the ratios of the protein contents in membrane and cytosol fractions upon differential centrifugation by Western blots[66]. Antibodies (BD Biosciences, catalogue Nr. 610090 and 610086) were diluted by 1:1000 for use. Total c-Jun N-terminal kinase (tJNK) and Thr[183]/Tyr[185]-phosphorylated JNK were quantified using specific antibodies (Cell Signaling Technology, catalogue Nr. 9252 and 9255, dilution 1:1000). Protein of Mfn2 (Abcam, catalogue Nr. ab56889, dilution 1:1000), Opa1 (BD Biosciences, catalogue Nr. 612607, dilution 1:1000), Fis1 (Merckmillipore, catalogue Nr. ABC67, dilution 1:1000), LC3 (Cell Signaling technology, catalogue Nr. 4108, dilution 1:1000), p62 (BD Biosciences, catalogue Nr. 610833, dilution 1:1000), DRP1 (Cell Signaling technology, catalogue Nr. 5391, dilution 1:1000), phospho-Ser[616]-DRP1 (Cell Signaling technology, catalogue Nr. 3455, dilution 1:1000), Pink1 (Abcam, catalogue Nr. ab23707, dilution 1:1000), phospho-Thr[257]-Pink1 (Ubiquigent, catalogue Nr. 68-0057-100, dilution 1:200), Parkin (Abcam, catalogue Nr. ab15954, dilution 1:1000), phospho-Ser[65]-Parkin (Ubiquigent, catalogue Nr. 68-0056-100, dilution 1:200), ETC Complex I-V (NADH:ubiquinone oxidoreductase subunit B8, succinate dehydrogenase complex iron sulfur subunit B, ubiquinol-cytochrome C reductase core protein 2, cytochrome c oxidase subunit IV, ATP synthase F1 subunit alpha, antibodies from Abcam, catalogue Nr. ab110413, dilution 1:300) were quantified in total protein lysates and were normalized to GADPH (Cell Signaling technology, catalogue Nr. 2118, dilution 1:20,000) as a loading control.

**Targeted lipidomics**. For quantification of lipid intermediates in subcellular muscle fractions, lipids were extracted, purified and analyzed from frozen tissue samples, using lipid chromatography mass spectrometry (LC-MS/MS)[67]. In brief, 50 mg of tissue was homogenized in 20 mM Tris/HCl, 1 mM EDTA 0.25 mM EGTA, pH 7.4, using a IKA T10 basic Ultra Turrax (IKA; Wilmington, NC, USA) and a tight-fitting glass douncer (Wheaton, Rochdale, UK). An internal standard (d517:0-DAG; Avanti Polar Lipids, Ala, USA) was added and samples were centrifuged for 1 h (100,000 × g, 4 °C). Lipid droplet, cytosol and membrane fractions were collected and lipids of each fraction were extracted according to Folch et al.[68], followed by solid phase extraction (Sep Pak Diol Cartridges; Waters, Milford, MA, USA). The resulting lipid phase was dried under a gentle flow of nitrogen and re-suspended in methanol. Lipid analytes were separated using a Phenomenex Luna Omega column (1.6 μm 100A; Phenomenex, Torrance, CA, USA) on an Infinity 1290 HPLC system (Agilent Technologies, Santa Clara, CA, USA) and analyzed by multiple reaction monitoring on a triplequadrupole mass spectrometer (Agilent 6495; Agilent Technologies), operated in the positive ion mode.

**Gene expression analyses**. Total RNA was extracted from 5–10 mg of muscle biopsies from the same participants described above using miRNA micro kit (Qiagen, Hilden, Germany) as per the manufacturer's protocol, with additional DNase treatment. All RNA samples with RNA integrity number RIN ≥7 (Bioanalyser, Agilent Technologies, Germany), were selected for microarray analysis. Transcriptome analysis was carried out by Oaklabs (Berlin, Germany) on their experimentally validated ArrayXS Human (design ID 079407, Agilent 60-mer SurePrint technology, Agilent Technologies). Gene ontology analysis was performed using David data base tools[69], with cutoff enrichment score set above 1.7 and enriched p value < 0.05 (Fisher test).

**DNA methylation**. Genomic DNA was extracted from 5 mg of skeletal muscle biopsies using Invisorb® Genomic DNA Kit II according to the manufacturer's protocol. An amount of 500 ng of genomic DNA from each participant was bisulfite-converted using Zymo EZ DNA Methylation-Gold kit (Zymo Research Corporation, Irvine, CA, USA) and then hybridized on Infinium®

MethylationEPIC BeadChip, (Eurofins Genomics GmbH, Ebersberg, Germany). EPIC chip covered 890,703 cytosine positions located in TSS200, TSS1500, 5UTR, 1Exon, gene body, 3UTR and intergenic regions of the human genome. Pre-processing and normalization included steps of probe filtering, color bias correction, background subtraction and subset quantile normalization and was processed with "ChAMP" background as previously described[70,71]. Mean change in beta methylation was calculated for a specific CpG site with comparing mean of beta methylation before to those at 2 or 52 weeks after surgery. As our study included both male and female participants, probes annotated to chromosome Y and X were excluded. All probes annotated to contain SNPs were excluded for unpaired analysis, but included for all paired analysis.

**Statistical analyses**. Normally distributed parameters are presented as means ± SD or means ± SEM, otherwise as median (interquartile range [IQR]). Not-normally distributed data were $log_e$-transformed to achieve near-normal distribution. Statistical analyses using covariance pattern model for repeated measures analysis based on patients with baseline and at least one follow-up muscle biopsy were performed. Analyses of the whole obese cohort were adjusted for age, sex, body mass index, surgery type and diabetes status at baseline and performed using SAS (version 9.4; SAS Institute, Cary, NC, USA). Phenotype traits were correlated to gene expression and DNA methylation by Pearson. Heat maps were generated with ggplots, R-package version 1.2.067. Programming and calculation of DNA methylation and transcripts were performed with R version 3.4.1 (2017-06-30). P values for transcriptome and methylome analysis were calculated by two-tailed Welch's t-test. P values of DNA methylation analysis were corrected by Benjamini Hochberg. Correction for multiple testing was performed for the methylome data and not for the transcriptome data. The reason for this is to avoid the number of false negatives and oversee relevant effects according to suggestions of John H. McDonald (McDonald, J.H. 2014. Handbook of Biological Statistics (3rd ed.). Sparky House Publishing, Baltimore, Maryland; p. 254–260). However, we provided the results of the multiple correction in the Supplementary Data 1 from A–E. A 4-field chi-square test was used for the enrichment analyses.

**Reporting summary**. Further information on research design is available in the Nature Research Reporting Summary linked to this article.

## Data availability

The metabolic dataset and all relevant western blots analyzed and reported in this study are included in the Source Data file. All array and methylation data analyzed and reported in this work have been deposited and are available at: (accession number GSE135066). R scripts used in this study have been uploaded to https://git.connect.dzd-ev.de/markusjaehnert/pmid_31519890. Supplementary Data 1A-E are available at https://git.connect.dzd-ev.de/markusjaehnert/pmid_31519890. All data are available from the corresponding author upon reasonable request.

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

## Acknowledgements

The authors would like to thank Kai Tinnes, Myrko Esser, Ilka Rokitta, David Höhn, Fariba Zivehe, and Andrea Sparla from the Institute for Clinical Diabetology, German Diabetes Center Dusseldorf for their excellent help with the experiments. This study was supported in part by the Ministry of Culture and Science of the State of North Rhine-Westfalia (MKW NRW), the German Federal Ministry of Health (BMG), by a grant of the Federal Ministry for Research (BMBF) to the German Center for Diabetes Research (DZD e.V., DZD Grant 2016), and by grants from the Helmholtz portfolio theme: Metabolic Dysfunction and Common Disease and the Helmholtz Alliance to Universities: Imaging and Curing Environmental Metabolic Diseases (ICEMED), the German Research Foundation (DFG, SFB 1116), German Diabetes Association (DDG) and the Schmutzler Stiftung.

## Author Contributions

S.G. performed the clinical experiments, analyzed data and wrote, edited, and reviewed the manuscript. M.O. performed gene expression and methylation analyses, analyzed data and wrote, edited, and reviewed the manuscript. C.K. and J.S. performed clinical experiments and edited and reviewed the manuscript. D.M. performed lipidomic analyses and edited and reviewed the manuscript. M.J and K.S. performed statistics analyses and edited and reviewed the manuscript. J.W. and F.G.S.T. performed electron microscopy analysis and edited and reviewed the manuscript. T.J., E.F. D.H.P., L.M., and C.H. performed laboratory analyses and edited and reviewed the manuscript. M.S. performed bariatric surgery procedures and edited and reviewed the manuscript. A.S. designed and led the gene transcription and methylation analysis and wrote, reviewed, and edited the manuscript. M.R. initiated the investigation, designed and led the clinical experiments, and wrote, reviewed, and edited the manuscript. All authors gave final approval of the version to be published. M.R. is the guarantor of this work and, as such, had full access to all the data in the study and takes responsibility for the integrity of the data and the accuracy of the data analysis.

## Additional information

**Competing interests:** The authors declare no competing interests.

