## [Peer Review File · Nature Communications]

Reviewers' comments:

Reviewer #1 (Remarks to the Author):

This is an interesting and well-written paper based on a study in humans that was designed to provide new insights into the mechanisms underlying improvements in insulin sensitivity following bariatric surgery. This is a topic of great interest, particularly in unravelling the effects that prevail acutely compared from those that occur later in association with weight loss. Here, the authors performed measurements in a large number (N=49) of obese men and women before and after (up to 52 weeks) bariatric surgery and healthy controls. Skeletal muscle biopsies were obtained for measuring mitochondria, targeted lipidomics, gene expression, DNA methylation, and protein expression. The goal here was to roll out these technologies to determine the association between specific molecular and cellular events in skeletal muscle with changes in peripheral insulin sensitivity. The main conclusion is that the improvements in insulin sensitivity observed 52 week following bariatric surgery are associated with some epigenetic changes in muscle that influence the expression of several genes that may be involved in metabolic changes. They propose an additional hypothesis that lipotoxicity and mitochondrial changes in the early timeframe post-surgery may precipitate subsequent epigenetic mechanisms. Despite what appears to be a very carefully conducted study with careful phenotyping and mechanistic information, there are some major considerations that should be addressed:

1. In many parts of the manuscript, the authors inappropriately ascribe causality between two variables. For example; in the 3rd sentence of the abstract, the authors state that initial weight loss increased muscle oxidative capacity. This may or may not be true. All we can say is that muscle oxidative capacity increased following surgery. Whether or not this was due to weight loss or simply paraphenomena cannot be determined with this study design. Similarly, one of the main conclusions in the abstract is that the early elevations on lipid intermediates from runaway lipolysis prevents rapid changes in insulin sensitivity post-surgery. This is not an unreasonable hypothesis, but such conclusive remarks seem overstated from the data available. Similarly, the idea that a transient increase in muscle oxidative capacity could induce epigenetic changes is interesting, but such causal inference in the current study seems a bit out of reach.
2. Skeletal muscle is probably the main contributor to insulin-stimulated glucose disposal during the hyperinsulinemic-euglycaemic clamp, but that somewhat depends on the levels of hyperinsulinemia achieved during the clamp. The methods are sparse on details, including how much insulin was infused and the prevailing insulin levels over the 3 hour period.
3. 10 lean controls were studied as a comparison group for the obese. The oxygen consumption data from permeabilized muscle fibres did not reach statistical significance despite what appears to be a meaningful magnitude of difference between the two groups. This is most likely due to what appears to be tremendous variability in the control group. In many cases, such as State U (Figure 2b,c), the differences between lean and obese are larger than the prospective change post-surgery. There is some confusion about the difference between figures 2b and 2c. Both are labeled state U, but the text on page 6 refers to 2b as state 3. Clarification is needed here.
4. If figure 2b is, indeed, state 3, then it is quite interesting that there was an acute decrease in state 3 (fully coupled respiration) yet an increase in state U. This needs some additional attention, such as limitation of adenine nucleotide translocase, etc.
5. Clamps were performed at 12 weeks and 24 weeks in addition to the other time points, as were mitochondrial respiration measurements. However, other outcomes are conspicuously absent at these intermediate timepoints, including DAGs, Ceramides and western blot markers, and transcriptomics. The conclusions would be greatly strengthened by including these additional intermediate timepoints for many outcomes.
6. In several instances, the authors refer to or make conclusion about data that is nowhere to be found in the manuscript or supplemental materials:
 - i. Please provide data on OPA1
 - ii. It is advisable to also look at mitochondrial fusion markers to gain a full perspective on mitochondrial dynamics
 - iii. Autophagy markers LC3 and p62 are not readily found
 - iv. In all cases, full representative western blots should be provided at least in the supplement for all protein targets.
 - v. PINK1, PARKIN, and DRP data should be shown.

7. The authors conclude that muscle mitochondrial content was lower at 2 weeks. This is based on measurements of citrate synthase activity and what appears to be a very preliminary analysis by electron microscopy. I question the validity of this conclusion based on N=2 for EM analysis. If the authors have muscle lysate available, they could blot for representative cytochrome chain proteins to better evaluate mitochondrial abundance.
8. Two genes were identified at 2 weeks that were interpreted as being involved in mitochondrial function. One is a tumor suppressor protein and the other is a methyltransferase. How are these related to mitochondrial function?
9. Supplemental experiments in C2C12 cells were performed to validate the interpretation that the expression of several salient muscle genes were influenced by the lipotoxic milieu at the early stages following surgery. The data seem to invalidate the conclusion. That is, high glucose appears to influence gene expression much more than palmitate. The value of these experiments is questionable, particularly since only palmitate was used as a fatty acid.

Minor comments:

1. Abstract, line 45: "...prevents from rapid changes..." This is awkward grammar.
2. Were the pathway analyses agnostic?

Reviewer #2 (Remarks to the Author):

The authors aimed to investigate the mechanisms underlying improvements in insulin sensitivity after bariatric surgery. Obese patients were analysed before and after bariatric surgery (N=49). Two weeks after surgery no changes in insulin sensitivity or inflammation were observed despite average 10kg weight loss. The authors describe amplified lipolysis of adipose tissue increasing circulating free fatty acids by 56% preventing rapid changes in insulin sensitivity early on. The authors describe immediate (2 weeks) altered regulation of genes involved in calcium/lipid metabolism and mitochondrial function. Furthermore these changes were associated with epigenetic modifications at 1 year where an average of 33% of weight loss was observed. Proinflammatory cytokine IL-6 and CRP reduced significantly and changes coincided with improvements in insulin sensitivity. Indeed pJNK/tJNK was lower at 1 year.

This paper sheds light on the trajectory of changes in insulin sensitivity post bariatric surgery. Specifically the authors describe novel results regarding genomic, transcriptomic and epigenomic changes that are associated with favourable changes in both calcium/lipid metabolism and mitochondrial function that serve to improve muscle energy metabolism and long term improvement in insulin sensitivity. Interestingly these epigenomic changes are shown in this research to favourable alter genetic expression of glucose and lipid metabolism.

Criticisms

1. Line 104 & 256: Express weight loss at 2 weeks as percentage weight loss
2. Body composition was not analysed at any time point. It is relevant to consider how changes in both fat mass and lean mass relate to the results described in this research. Morbidly obese patients such as those in this study commonly present with growth hormone deficiency which can significantly impact both retention of skeletal muscle in response to surgery and also improvements in insulin sensitivity (Savastano et al. 2009).
3. It is not detailed whether all subjects were diabetic. Please provide more detail.
4. Medications of subjects are not described. Please provide more detail and explain how this may have influenced the results.
5. Table 1: Add body weight and detail changes at each time point as both absolute and % changes

Reference:

Savastano, S., Di Somma, C., Angrisani, L., Orio, F., Longobardi, S., Lombardi, G. and Colao, A.,

2009. Growth hormone treatment prevents loss of lean mass after bariatric surgery in morbidly obese patients: results of a pilot, open, prospective, randomized, controlled study. *The Journal of Clinical Endocrinology & Metabolism*, 94(3), pp.817-826.

Carel le Roux

Reviewer #3 (Remarks to the Author):

Gancheva and colleagues report on gene expression and DNA methylation in muscle tissue from individuals prior to and after metabolic surgery. This is a very comprehensive clinical and translational study. Unfortunately, it is difficult for the reader to discern the biologically meaningful changes for several reasons. Firstly, presentation of pathway enrichment, without presenting directionality of change for either the pathway or individual genes within the pathway, makes it difficult for the reader to understand the net biological impact. Additional heat maps for selected pathways may be helpful, with expanded discussion for top-ranking pathways highlighting predicted impact on physiology/metabolism. Secondly, the authors make comments about causal relationships between expression and DNA methylation-dependent regulation which cannot be assessed in this longitudinal but descriptive analysis. These conclusions need to be modified to reflect associations. Finally, it would be great if the authors could relate the differential expression to the primary metabolic phenotypes of interest (e.g. weight, insulin sensitivity) to take advantage of their unique time course data, with interindividual differences. Which alterations were associated with change in insulin sensitivity for an individual patient, assessed over time?

Major:

Please provide information about gene expression and methylation profiles which paralleled changes in BMI or changes in insulin sensitivity or differences in delta RQ. This would allow the authors to identify in an unbiased way, weight-dependent vs. independent changes in expression/methylation. As it stands, the authors have focused on longitudinal changes in mitochondrial function, lipid metabolism, and calcium signaling only, based on per timepoint comparative analysis.

Line 230 - The authors indicate they tested whether epigenetic alterations were responsible for this effect, analyzing differentially expressed genes at 2 wks in parallel with methylation at 52 weeks. Cause and effect relationships cannot be discerned from this analysis. Moreover the statement in line 238-240 is not valid – changes in methylation at 52 weeks do not necessarily participate in reversal of gene expression from 2 to 52 weeks.

Similarly, lines 263-265 – the authors indicate that changes in endogenous FFA induce with DAG/PKC pathway. Associations can be noted, but causality cannot be concluded. These conclusions need to be revised to indicate association.

Similarly, lines 360-362 – the conclusion that “changes in DNA methylation reprogram up to 70% of the transiently altered transcripts...” implies causality which cannot be discerned.

Can the authors comment on how much of the time-dependent variation may be related to differences related to repeated measures? Changes in diet/activity?

Line 563 – Was the % of transcripts without differential expression at 2 weeks that had changes in DNA methylation at 52 weeks? How much different was the pattern in those with differential expression at 2 weeks?

Please comment on biological significance of relatively low changes in methylation (either absolute or relative) over time. Despite statistical significance, the low magnitude of changes are unlikely to be major contributors to altered gene expression.

Minor:

1. Did results differ according to type of surgery (RYGB vs. SG)?
2. OBE is an unusual abbreviation for obese. Consider OB or including full word.
3. Figure 1 – please reorder figure panels to be consistent with time first mentioned in text.
4. Line 88 – what do the authors mean about gradually higher for PKC?
5. Line 187 – what does the term “variation of DNA methylation” mean? Referring to change in

methylation?

6. Line 250 – what does continuous mean in this context?

7. Line 482 – please clarify how p values were adjusted for multiple comparisons for the transcriptome and metabolomic analysis.

8. Figure 3 – were the heat maps row-normalized?

Point-by point replies to the reviewers' comments

Reviewer #1

This is an interesting and well-written paper based on a study in humans that was designed to provide new insights into the mechanisms underlying improvements in insulin sensitivity following bariatric surgery. This is a topic of great interest, particularly in unravelling the effects that prevail acutely compared from those that occur later in association with weight loss. Here, the authors performed measurements in a large number (N=49) of obese men and women before and after (up to 52 weeks) bariatric surgery and healthy controls. Skeletal muscle biopsies were obtained for measuring mitochondria, targeted lipidomics, gene expression, DNA methylation, and protein expression. The goal here was to roll out these technologies to determine the association between specific molecular and cellular events in skeletal muscle with changes in peripheral insulin sensitivity. The main conclusion is that the improvements in insulin sensitivity observed 52 week following bariatric surgery are associated with some epigenetic changes in muscle that influence the expression of several genes that may be involved in metabolic changes. They propose an additional hypothesis that lipotoxicity and mitochondrial changes in the early timeframe post-surgery may precipitate subsequent epigenetic mechanisms. Despite what appears to be a very carefully conducted study with careful phenotyping and mechanistic information, there are some major considerations that should be addressed:

- *We would like to thank the reviewer for the overall rating of our study as being interesting, very carefully conducted and addressing a topic of great interest. We also appreciated the constructive considerations, which we aimed to address as follows.*

1. In many parts of the manuscript, the authors inappropriately ascribe causality between two variables. For example; in the 3rd sentence of the abstract, the authors state that initial weight loss increased muscle oxidative capacity. This may or may not be true. All we can say is that muscle oxidative capacity increased following surgery. Whether or not this was due to weight loss or simply parphenomena cannot be determined with this study design. Similarly, one of the main conclusions in the abstract is that the early elevations on lipid intermediates from runaway lipolysis prevents rapid changes in insulin sensitivity post-surgery. This is not an unreasonable hypothesis, but such conclusive remarks seem overstated from the data available. Similarly, the idea that a transient increase in muscle oxidative capacity could induce epigenetic changes is interesting, but such causal inference in the current study seems a bit out of reach.

- *We appreciated these comments and therefore changed the wording accordingly to imply association and not causality in the respective parts of the Abstract and Discussion (p. 3, lines 46-49, p. 13 lines 302-309). We have also down tuned the statements on induction of epigenetic changes by altered muscle oxidative capacity. Nevertheless, some recent reports provided supporting information on the role of in oxidative capacity for epigenetic changes¹⁻³. This aspect has been addressed in the discussion (p. 17, lines 417-418).*

2. Skeletal muscle is probably the main contributor to insulin-stimulated glucose disposal during the hyperinsulinemic-euglycaemic clamp, but that somewhat depends on the levels of hyperinsulinemia

achieved during the clamp. The methods are sparse on details, including how much insulin was infused and the prevailing insulin levels over the 3 hour period.

- *We apologize for the lack of detail on the methods. Participants received a primed continuous infusion of 40 mU of human regular insulin per m², of body surface area per min, as used and described in our previous papers⁴(e. g. Koliaki et al. Cell Metab 2015,21:739-746). This infusion rate leads stable serum insulin concentrations of 58±14 μU/ml during the hyperinsulinemic clamp steady state. This is now reported in the Methods (p. 19, lines 476-482) and Results (p. 5, lines 85-86) section of the manuscript.*

3. 10 lean controls were studied as a comparison group for the obese. The oxygen consumption data from permeabilized muscle fibres did not reach statistical significance despite what appears to be a meaningful magnitude of difference between the two groups. This is most likely due to what appears to be tremendous variability in the control group. In many cases, such as State U (Figure 2b,c), the differences between lean and obese are larger than the prospective change post-surgery.

- *We thank the reviewer for this helpful comment. In order to clarify this issue, we included another 7 healthy humans into the control group. The updated analysis revealed lower variability and statistically significant differences in uncoupled mitochondrial respiration and respiratory control ratio (RCR) between the groups (Results section: p. 5, line 98-103). This further confirmed the notion of altered muscle mitochondrial function in obesity which subsequently changes following bariatric surgery.*

There is some confusion about the difference between figures 2b and 2c. Both are labeled state U, but the text on page 6 refers to 2b as state 3. Clarification is needed here.

- *Both former Figures 2b and 2c reported state U respiration, however, once expressed per mg muscle tissue and once, per CSA. State 3 respiration per mg liver tissue was only reported in the text, but showed the same direction of changes as state U expressed per mg muscle tissue (decrease by 16 % and 11 %, respectively). Comparison of O₂ flux rates per mg tissue and complex II showed either decrease or no change at 2 weeks after surgery. This has now been clarified in the revised version (p .6, lines 129-130).*

4. If figure 2b is, indeed, state 3, then it is quite interesting that there was an acute decrease in state 3 (fully coupled respiration) yet an increase in state U. This needs some additional attention, such as limitation of adenine nucleotide translocase, etc.

- *The former Figure 2b (now figure 2a) showed changes in state U respiration and similar time course of changes has been observed for state 3 as well (see below).*

5. Clamps were performed at 12 weeks and 24 weeks in addition to the other time points, as were

mitochondrial respiration measurements. However, other outcomes are conspicuously absent at these intermediate timepoints, including DAGs, Ceramides and western blot markers, and transcriptomics. The conclusions would be greatly strengthened by including these additional intermediate timepoints for many outcomes.

- *As these measurements are quite laborious and we originally did not expect relevant information, we only performed it now in response to this comment:*
 - (i) *These measurements of DAG in different compartments, ceramides and Western blot analyses for the timepoints 12 and 24 week revealed increases in individual DAG species at 12 and 24 weeks, specifically in the membrane and lipid droplet fractions (Suppl. fig. 2). This was paralleled by the only gradual lowering of PKC δ activation at these time points (Figure 1c). These findings now suggest the lipotoxic effect persists at 12 and 24 weeks after surgery, possibly contributing to the slow recovery of peripheral insulin sensitivity by 25 % and 46 % at 12 and 24 weeks, respectively. Only at 52 weeks after surgery, obese participants achieved the degree of insulin sensitivity typical for healthy humans. Of note, the better parameter of adipose tissue insulin resistance, i. e. Adipo-IR (calculated from fasting FFA and fasting insulin levels) - when compared with FFA disappearance during the hyperinsulinemic clamp, clearly shows that FFA flux remained elevated for long time along with the lower whole-body (muscle insulin sensitivity). We therefore decided to present the Adipo-IR instead of the FFA disappearance during the clamp in the revised Fig. 1 a. We now include these potentially interesting, further insights into the time course of post-surgical metabolic changes in the revised manuscript.*
 - (ii) *Furthermore, we found stable levels of Mitofusin-1, Drp1 and Opa1 at 12 and 24 weeks and additionally assessed Fis1 content as a marker of mitochondrial fission, which showed no changes at 2, 12 and 24 weeks, but rose at 52 weeks.*
 - (iii) *In addition, we performed transcriptome analysis for the time points 12 and 24 weeks. We included the number of differentially expressed genes in Fig. 3a. We performed pathway enrichment analysis which we show in Suppl. fig. 5b and c and which we described in the results (p. 9, line 201-208). Furthermore, we provide expression data of the candidates (TBC1D1, ASPSCR1, NR4A1, and ELOVL5 in Fig. 5 as well as for FTO and TOMM7 of Fig. 6) for the entire time course in Suppl. fig. 9 and describe the results on p. 11, lines 265-267.*

6. In several instances, the authors refer to or make conclusion about data that is nowhere to be found in the manuscript or supplemental materials:

i. Please provide data on OPA1

- *We are sorry for not providing information on the OPA1 data, which are now included in Suppl. fig. 3.*

ii. It is advisable to also look at mitochondrial fusion markers to gain a full perspective on mitochondrial dynamics

- *We now show all available data on mitochondrial fusion and fission markers Mfn2, Opa1, Drp1 and Fis1 at all time points after surgery (Suppl. fig. 3).*

iii. Autophagy markers LC3 and p62 are not readily found

- *We now also provide information on LC3 and p62 on p. 5, lines 107-108, p. 6, lines 139-140, p. 7, lines 163-164.*

iv. In all cases, full representative western blots should be provided at least in the supplement for all protein targets.

- *This has now been included in Suppl. fig. 11.*

v. PINK1, PARKIN, and DRP data should be shown.

- *This data has now been included in Suppl. fig 3.*

7. The authors conclude that muscle mitochondrial content was lower at 2 weeks. This is based on measurements of citrate synthase activity and what appears to be a very preliminary analysis by electron microscopy. I question the validity of this conclusion based on N=2 for EM analysis. If the authors have muscle lysate available, they could blot for representative cytochrome chain proteins to better evaluate mitochondrial abundance.

- *We agree with the reviewer that such conclusions should not be based quantitatively on one parameter. We now performed Western blot analysis of all mitochondrial electron transport chain complexes (ETC) (Suppl. fig. 2, Fig. 2c). The data reveal a mixed picture with lower levels of CII and CIII in the obese at baseline, trends towards lower levels of CII and higher levels of CV at 2 weeks after surgery (Suppl. fig. 2). These data may suggest differences in the adaptation of mitochondrial mass resulting in variable oxidative capacity. Nevertheless, the respiratory control ratio (RCR), which reflects mitochondrial efficacy and is not depending on any measure of mitochondrial mass, is lower in obese at baseline and decreases shortly after surgery, suggesting impairment of mitochondrial function independent of mitochondrial content. We now show these additional data and modified the Discussion accordingly (p.14 , lines 342-357).*

8. Two genes were identified at 2 weeks that were interpreted as being involved in mitochondrial function. One is a tumor suppressor protein and the other is a methyltransferase. How are these related to mitochondrial function?

- *MTUS1 is a microtubule-associated tumor suppressor which is located in mitochondria and has recently been implicated in mitochondrial function and morphology⁵. TRMT6 is a methyltransferase, which is dominantly localized in mitochondria and known to modify human mitochondrial tRNAs^{6,7}. These links to mitochondrial function have now been also included into the manuscript (p. 16, lines 397-400, p9, line 195)*

9. Supplemental experiments in C2C12 cells were performed to validate the interpretation that the expression of several salient muscle genes were influenced by the lipotoxic milieu at the early stages following surgery. The data seem to invalidate the conclusion. That is, high glucose appears to influence gene expression much more than palmitate. The value of these experiments is questionable, particularly since only palmitate was used as a fatty acid.

- *We thank the reviewer for pointing out this issue and followed the suggestion to remove this data.*

Minor comments:

1. Abstract, line 45: "...prevents from rapid changes..." This is awkward grammar.

- *We changed this phrase to “lack of rapid changes” (p. 3, line 49)*

2. Were the pathway analyses agnostic?

- *The pathway analysis shown in Figure 3b-c and 4b are not agnostic, all genes differentially expressed were used for gene ontology analysis and the results are depicted in GO circle plots. However, data shown in Figure 3d on mitochondria-, lipid- and calcium-related genes were agnostic because they appeared in the top 10 upregulated genes at 2 weeks.*

Reviewer #2

The authors aimed to investigate the mechanisms underlying improvements in insulin sensitivity after bariatric surgery. Obese patients were analysed before and after bariatric surgery (N=49). Two weeks after surgery no changes in insulin sensitivity or inflammation were observed despite average 10kg weight loss. The authors describe amplified lipolysis of adipose tissue increasing circulating free fatty acids by 56% preventing rapid changes in insulin sensitivity early on. The authors describe immediate (2 weeks) altered regulation of genes involved in calcium/lipid metabolism and mitochondrial function. Furthermore these changes were associated with epigenetic modifications at 1 year where an average of 33% of weight loss was observed. Proinflammatory cytokine IL-6 and CRP reduced significantly and changes coincided with improvements in insulin sensitivity. Indeed pJNK/tJNK was lower at 1 year.

This paper sheds light on the trajectory of changes in insulin sensitivity post bariatric surgery. Specifically the authors describe novel results regarding genomic, transcriptomic and epigenomic changes that are associated with favourable changes in both calcium/lipid metabolism and mitochondrial function that serve to improve muscle energy metabolism and long term improvement in insulin sensitivity. Interestingly these epigenomic changes are shown in this research to favourable alter genetic expression of glucose and lipid metabolism.

- *We would like to thank the reviewer for acknowledging the novelty of genomic, transcriptomic and epigenomic changes. We also appreciated the constructive criticism, which we aimed to address as follows.*

Criticisms

1. Line 104 & 256: Express weight loss at 2 weeks as percentage weight loss

- *We now give the percent change in body weight in the text and in Table 1 (p. 6, line 116, p. 7, line 147 and Table 1).*

2. Body composition was not analysed at any time point. It is relevant to consider how changes in both fat mass and lean mass relate to the results described in this research. Morbidly obese patients such as those in this study commonly present with growth hormone deficiency which can significantly impact both retention of skeletal muscle in response to surgery and also improvements in insulin sensitivity (Savastano et al. 2009).

We agree with the reviewer that changes in lean and fat mass are relevant for the improvement of insulin sensitivity after bariatric surgery, as shown before. In the absence of a direct measure of lean/fat mass in this study, we now measured serum leptin levels, which decreased after surgery in line with previous reports^{8,9} and is known to tightly relate to fat mass¹⁰. In order to address the question on growth hormone in obesity (Savastano, S., Di Somma, C., Angrisani, L., Orio, F., Longobardi, S., Lombardi, G. and Colao, A., 2009. Growth hormone treatment prevents loss of lean mass after bariatric surgery in morbidly obese patients: results of a pilot, open, prospective, randomized, controlled study. The Journal of Clinical Endocrinology & Metabolism, 94(3), pp.817-826), we now also measured growth hormone and IGF-1 levels. Interestingly, our cohort showed no evidence of growth hormone deficiency. Please find below a summary of the detailed results, which are not included in the manuscript due to space limitation (p. 19, line 470-471).

	CON	OB before surgery	OB 52 weeks after surgery
Leptin (ng/ml)	5.9±4.5	72.0±26.0 [#]	25.5±10.9 [#]
Growth hormone (pg/ml)	737±1218	642±751	2809±2924 [#]
IGF-1 (ng/ml)	108±29	95±43	117±29

Data are mean±SD, [#]p<0.05 vs CON

3. It is not detailed whether all subjects were diabetic. Please provide more detail.

- *Thirteen of the obese participants had type 2 diabetes at baseline, but exhibited very good metabolic control with HbA1c of 7.2±1.0% and fasting blood glucose of 139±24 mg/dl. We now provide further details on p. 17, lines 414-423.*

4. Medications of subjects are not described. Please provide more detail and explain how this may have influenced the results.

- *Our study allows participants to be on stable doses of thyroid hormone replacement, antihypertensive and/or oral contraceptive therapy, participants taking other relevant medications including any immunomodulatory medications were excluded, which has been now added to the Methods (p. 18-19, lines 458-467).*

5. Table 1: Add body weight and detail changes at each time point as both absolute and % changes

- *This information is now included in Table 1.*

Reviewer #3

Gancheva and colleagues report on gene expression and DNA methylation in muscle tissue from individuals prior to and after metabolic surgery. This is a very comprehensive clinical and translational study.

- *We would like to thank the reviewer for describing our work as a very comprehensive clinical and translational study. We also appreciated the constructive comments, which we aimed to address as follows.*

Unfortunately, it is difficult for the reader to discern the biologically meaningful changes for several reasons. Firstly, presentation of pathway enrichment, without presenting directionality of change for either the pathway or individual genes within the pathway, makes it difficult for the reader to understand the net biological impact. Additional heat maps for selected pathways may be helpful, with expanded discussion for top-ranking pathways highlighting predicted impact on physiology/metabolism.

- *We agree with the reviewer that it was difficult to estimate the outcome of the pathway analysis. Thus, we now show the results of the pathway analysis as circular visualization of gene-annotation enrichment analysis, which provides the direction and levels of changes, and the number of genes that are affected (Fig. 3b, Fig. 4b). Accordingly, additional heat maps would be redundant.*

Secondly, the authors make comments about causal relationships between expression and DNA methylation-dependent regulation which cannot be assessed in this longitudinal but descriptive analysis. These conclusions need to be modified to reflect associations.

- *We fully agree with the reviewer and adapted and down tuned our discussion accordingly (p. 10, lines 240, p. 9, line 217-218, p. 12, line 295, p. 13, line 307, p. 17, lines 429, p. 18, line 434-435, p. 18, line 438-439 and p. 18, line 447-449).*

Finally, it would be great if the authors could relate the differential expression to the primary metabolic phenotypes of interest (e.g. weight, insulin sensitivity) to take advantage of their unique time course data, with interindividual differences. Which alterations were associated with change in insulin sensitivity for an individual patient, assessed over time?

- *We very much appreciate this comment. In response, we first performed additional measurements at the intermediate time points (12 and 24 weeks) and performed comprehensive correlation analyses. The new Table 2 summarizes the numbers of differentially expressed genes (Table 2a), and of differentially expressed and methylated genes (Table 2b), which can be linked to the different phenotypes. We also provide a list of the corresponding genes in Suppl. Table 10 and refer to these data in the revised manuscript (p. 11/12, lines 269-281).*

Major:

Please provide information about gene expression and methylation profiles which paralleled changes in BMI or changes in insulin sensitivity or differences in delta RQ. This would allow the authors to identify in an unbiased way, weight-dependent vs. independent changes in expression/methylation. As it stands, the authors have focused on longitudinal changes in mitochondrial function, lipid metabolism, and calcium signaling only, based on per timepoint comparative analysis.

- *We thank the reviewer for this comment. Our metabolic analyses were adjusted for baseline BMI and thereby reveal weight-independent changes. In addition, we performed Principal Component Analysis (PCA) for methylome as well as for transcriptome data with BMI (please see below). Importantly, these PCA - which do not show clusters- indicate that changes in expression and methylation are independent of body weight.*

PCA of DNA methylation before surgery (0 w) PCA of DNA methylation after surgery (52 w)

Line 230 - The authors indicate they tested whether epigenetic alterations were responsible for this effect, analyzing differentially expressed genes at 2 wks in parallel with methylation at 52 weeks. Cause and effect relationships cannot be discerned from this analysis. Moreover the statement in line 238-240 is not valid – changes in methylation at 52 weeks do not necessarily participate in reversal of gene expression from 2 to 52 weeks.

- *We agree and rephrased this part accordingly to express association and not causality (p. 12, lines 295).*

Similarly, lines 263-265 – the authors indicate that changes in endogenous FFA induce with DAG/PKC pathway. Associations can be noted, but causality cannot be concluded. These conclusions need to be revised to indicate association.

- *This has now been modified to reflect the association and not causality (p. 13, lines 323-324)*

Similarly, lines 360-362 – the conclusion that “changes in DNA methylation reprogram up to 70% of the transiently altered transcripts...” implies causality which cannot be discerned.

- *This has now been modified to read “are associated with reprogramming up to 70% of the transiently altered transcripts in order to possibly normalize their expression levels at 52 weeks”.*

Can the authors comment on how much of the time-dependent variation may be related to differences related to repeated measures? Changes in diet/activity?

- *We used the paired test as an appropriate test to remove the time-dependent variation of our analysis. Indeed diet and eating behavior are important determinants of insulin sensitivity, which might have influenced our results. Participants in the study were not enrolled in structured training or diet programs (as this was no intervention trial) so that related measurements/analyses were not performed, which is now described as a limitation (p. 15, lines 371-374).*

Line 563 – Was the % of transcripts without differential expression at 2 weeks that had changes in DNA methylation at 52 weeks? How much different was the pattern in those with differential expression at 2 weeks?

- *None of the remaining transcripts (27%), for which we detected an altered DNA methylation at 52 weeks exhibited a differential expression at 2 weeks.*

Please comment on biological significance of relatively low changes in methylation (either absolute or relative) over time. Despite statistical significance, the low magnitude of changes are unlikely to be major contributors to altered gene expression.

- *We acknowledge the relevance of this question, but have to state that it is difficult to comment on the less-well defined term “biological significance”. While the changes in DNA methylation may indeed appear to be small, this effect size is usually observed and described for metabolically relevant genes. Furthermore, the functional relevance of moderate differences in DNA methylation was proved by us for some genes by reporter assays^{11,12}. This is now briefly discussed on p.17, lines 413-417.*

Minor:

1. Did results differ according to type of surgery (RYGB vs. SG)?

- *We thank the reviewer for this interesting comment. A subgroup analysis of patients undergoing RYGB and SG revealed no differences in time course of weight loss and insulin sensitivity compared to the pooled group of all participants (please see below). Because this had not been the primary aim of study and due to the limited subgroup size and the allowed word count, we would prefer not to include this information in the manuscript.*

2. OBE is an unusual abbreviation for obese. Consider OB or including full word.

- *We now changed the abbreviation to OB in our revisions.*

3. Figure 1 – please reorder figure panels to be consistent with time first mentioned in text.

- *The panels of Figure 1 have been reordered according to sequence of mentioning in the text.*

4. Line 88 – what do the authors mean about gradually higher for PKC?

- *This has now been rephrased to read “there was a trend for higher PKC β ” in order to avoid misunderstanding and the respective p value is reported.*
5. Line 187 – what does the term “variation of DNA methylation” mean? Referring to change in methylation?
- *We refer to change in DNA methylation and corrected this phrase accordingly (p. 10, line 224 and 228, p.).*
6. Line 250 – what does continuous mean in this context?
- *We removed “continuous” to avoid any confusion (p. 13, line 307).*
7. Line 482 – please clarify how p values were adjusted for multiple comparisons for the transcriptome and metabolomic analysis.
- *We thank the reviewer for this comment. We believe that this comment meant multiple corrections instead of multiple comparison. Multiple correction was not applied in our analysis and as stated in Perneger et al¹³, these adjustments would not be appropriate for clinically-based studies.*
8. Figure 3 – were the heat maps row-normalized?
- *Our heat maps are row-normalized using the log-transformed magnitude of the average of expression per group (s. Figure 3 legend).*

References

1. Wallace DC, Fan W. Energetics, epigenetics, mitochondrial genetics. *Mitochondrion* 2010; **10**(1): 12-31.
2. Pietrocola F, Galluzzi L, Bravo-San Pedro JM, Madeo F, Kroemer G. Acetyl coenzyme A: a central metabolite and second messenger. *Cell Metab* 2015; **21**(6): 805-21.
3. Matilainen O, Quiros PM, Auwerx J. Mitochondria and Epigenetics - Crosstalk in Homeostasis and Stress. *Trends Cell Biol* 2017; **27**(6): 453-63.
4. Koliaki C, Szendroedi J, Kaul K, et al. Adaptation of hepatic mitochondrial function in humans with non-alcoholic fatty liver is lost in steatohepatitis. *Cell metabolism* 2015; **21**(5): 739-46.
5. Wang Y, Huang Y, Liu Y, et al. Microtubule associated tumor suppressor 1 interacts with mitofusins to regulate mitochondrial morphology in endothelial cells. *FASEB journal : official publication of the Federation of American Societies for Experimental Biology* 2018; **32**(8): 4504-18.
6. Chujo T, Suzuki T. Trmt61B is a methyltransferase responsible for 1-methyladenosine at position 58 of human mitochondrial tRNAs. *RNA (New York, NY)* 2012; **18**(12): 2269-76.
7. Safra M, Sas-Chen A, Nir R, et al. The m1A landscape on cytosolic and mitochondrial mRNA at single-base resolution. *Nature* 2017; **551**(7679): 251-5.
8. Khosravi-Largani M, Nojomi M, Aghili R, et al. Evaluation of all Types of Metabolic Bariatric Surgery and its Consequences: a Systematic Review and Meta-Analysis. *Obesity surgery* 2019; **29**(2): 651-90.

9. Wolf RM, Jaffe AE, Steele KE, et al. Cytokine, Chemokine, and Cytokine Receptor Changes Are Associated With Metabolic Improvements After Bariatric Surgery. *The Journal of clinical endocrinology and metabolism* 2019; **104**(3): 947-56.
10. Maffei M, Halaas J, Ravussin E, et al. Leptin levels in human and rodent: measurement of plasma leptin and ob RNA in obese and weight-reduced subjects. *Nature medicine* 1995; **1**(11): 1155-61.
11. Baumeier C, Saussenthaler S, Kammel A, et al. Hepatic DPP4 DNA Methylation Associates With Fatty Liver. *Diabetes* 2017; **66**(1): 25-35.
12. Kammel A, Saussenthaler S, Jahnert M, et al. Early hypermethylation of hepatic Igfbp2 results in its reduced expression preceding fatty liver in mice. *Human molecular genetics* 2016; **25**(12): 2588-99.
13. Perneger TV. What's wrong with Bonferroni adjustments. *BMJ* 1998; **316**(7139): 1236-8.

Reviewers' comments:

Reviewer #1 (Remarks to the Author):

The authors have responded to all of my initial queries and provided additional data and revisions to my satisfaction. I have no additional requests.

Reviewer #2 (Remarks to the Author):

I would like to thank the authors of taking the time and effort to respond appropriately to the comments made. I can confirm that all points have been addressed appropriately.

Reviewer #3 (Remarks to the Author):

The authors have robustly responded to the reviewer queries, generating a much improved analysis overall.

Minor points:

Adjustment for baseline BMI nor PCA plots of data colored by BMI demonstrate whether weight-independent changes contribute to observed phenotypes. This still should be a point of discussion.

Also, time dependent variation could be related to factors beyond the intervention. Paired t -test does not address this issue. If a patient not undergoing the intervention was resampled, how much variation would there be? This is not addressed.

Please mention the subset analysis of RYGB and SG - this is an important point noted in the reviewer response which should be noted in the manuscript.

Omics data should still be adjusted given the large numbers of analytes/transcripts being studied. Please clarify.

Replies to Reviewers' comments

Reviewer #1

The authors have responded to all of my initial queries and provided additional data and revisions to my satisfaction. I have no additional requests.

- *We would like to thank the reviewer for expressing satisfaction with our additional data and revisions.*

Reviewer #2

I would like to thank the authors of taking the time and effort to respond appropriately to the comments made. I can confirm that all points have been addressed appropriately.

- *We would like to thank the reviewer for the kind words and confirmation of the adequate response to the points raised.*

Reviewer #3

The authors have robustly responded to the reviewer queries, generating a much improved analysis overall.

- *We would like to thank the reviewer for rating of our response to the queries and our analysis of the data.*

Minor points:

Adjustment for baseline BMI nor PCA plots of data colored by BMI demonstrate whether weight-independent changes contribute to observed phenotypes. This still should be a point of discussion.

- *The reviewer is correct that we did not directly address whether and to which extent weight-independent changes contribute to the observed changes. The fact that the number of genes, which correlate with BMI ($n=17$) is lower than the number of genes, which correlate for instance with fasting blood glucose ($n=231$) or HMW-adiponectin ($n=61$) - shown in Table 2 - indicates that most effects occur independently of changes in body weight. We now refer to this point in the Discussion by stating: "The question is whether all changes in expression and DNA methylation are occurring in response to weight loss or if and to which extent weight-independent alterations contribute to observed phenotypes. The finding that the number of genes correlating with BMI is clearly lower than that of genes associating for instance with fasting glycemia or adiponectin levels indicates that not all observed effects are directly related to body weight loss."*

Also, time dependent variation could be related to factors beyond the intervention. Paired t-test does not address this issue. If a patient not undergoing the intervention was resampled, how much variation would there be? This is not addressed.

- *This point is now addressed in the Discussion section: "In addition to the changes resulting from surgery, moderate alterations of measured variables could be also due to the repeated testing one year later." As repeated analysis of the control group was not part of the study protocol and is therefore not available we cannot calculate the variation. We know from previous clinical studies that the metabolic parameters remain quite stable in healthy humans over the course of year provided weight maintenance." This is now discussed on p. 14 line 346-348.*

Please mention the subset analysis of RYGB and SG - this is an important point noted in the reviewer response which should be noted in the manuscript.

- *We would like reiterate our reply to the previous minor comment 1 of this reviewer that the group size does not allow to draw firm conclusions as to metabolic differences between both surgical procedures. To satisfy the reviewer, we decided to add a statement on possible similarities between the procedures: "A subgroup comparison between patients undergoing gastric bypass and sleeve gastrectomy surgery did not suggest differences in the time course of changes in body weight and muscle insulin sensitivity. However, the relatively small subgroup size does not allow to draw firm conclusions on a possible metabolic difference between these surgical techniques."*

Omics data should still be adjusted given the large numbers of analytes/transcripts being studied. Please clarify.

- *Adjusting data for multiple testing is in general advised to reduce the number of false positives. However, by correcting for multiple testing one would increase the number of false negatives and thereby miss some effects. Due to the rather small sample size, we would miss many effects. Thus, correction for multiple hypothesis testing was not performed following the suggestions of John H. McDonald in order not to increase the number of false negatives and miss important effects (McDonald, J.H. 2014. Handbook of Biological Statistics (3rd ed.). Sparky House Publishing, Baltimore, Maryland; p. 254-260).*

REVIEWERS' COMMENTS:

Reviewer #3 (Remarks to the Author):

Thank you for your modifications. I have no further questions.